# Cell lineage-dependent chiral actomyosin flows drive cellular rearrangements in early *Caenorhabditis elegans* development

Lokesh G Pimpale[1,2,3], Teije C Middelkoop[1,2,3], Alexander Mietke[1,4,5,6,7], Stephan W Grill[1,2,3]*

[1]Max Planck Institute of Molecular Cell Biology and Genetics, Dresden, Germany; [2]Biotechnology Center, TU Dresden, Dresden, Germany; [3]Cluster of Excellence Physics of Life, TU Dresden, Dresden, Germany; [4]Max Planck Institute for the Physics of Complex Systems, Dresden, Germany; [5]Chair of Scientific Computing for Systems Biology, Faculty of Computer Science, TU Dresden, Dresden, Germany; [6]Center for Systems Biology Dresden, Dresden, Germany; [7]Department of Mathematics, Massachusetts Institute of Technology, Cambridge, United States

**Abstract** Proper positioning of cells is essential for many aspects of development. Daughter cell positions can be specified via orienting the cell division axis during cytokinesis. Rotatory actomyosin flows during division have been implied in specifying and reorienting the cell division axis, but how general such reorientation events are, and how they are controlled, remains unclear. We followed the first nine divisions of *Caenorhabditis elegans* embryo development and demonstrate that chiral counter-rotating flows arise systematically in early AB lineage, but not in early P/EMS lineage cell divisions. Combining our experiments with thin film active chiral fluid theory we identify a mechanism by which chiral counter-rotating actomyosin flows arise in the AB lineage only, and show that they drive lineage-specific spindle skew and cell reorientation events. In conclusion, our work sheds light on the physical processes that underlie chiral morphogenesis in early development.

*For correspondence:
grill@mpi-cbg.de

**Competing interests:** The authors declare that no competing interests exist.

## Introduction

Proper positioning of cells is instrumental for many aspects of early embryonic development (*Wong et al., 2010*; *Bronner, 2012*; *Buchsbaum and Cappello, 2019*). One way to achieve proper cell positioning is via migration (*Scarpa and Mayor, 2016*; *De Pascalis and Etienne-Manneville, 2017*). Another way by which cell positions can be specified is via the orientation of the cell division axis (*Fishkind and Wang, 1995*; *Glotzer, 1997*; *Field et al., 1999*). Since the orientation of the mitotic spindle dictates the cell division axis, the orientation of the mitotic spindle at cleavage plays a key role in determining daughter cell positions (*Haydar et al., 2003*; *Egger et al., 2007*; *Kulukian and Fuchs, 2013*; *Howard and Garzon-Coral, 2017*). This axis can be set at the beginning of cytokinesis, by assembling the mitotic spindle in the correct orientation from the start, or during cytokinesis by reorientation of the mitotic spindle (*Wilson, 1925*; *Meshcheryakov and Beloussov, 1975*; *Kaltschmidt et al., 2000*; *Matsumura et al., 2012*; *Sugioka and Bowerman, 2018*; *Galli and van den Heuvel, 2008*). While several studies highlight the mechanisms that determine the initial orientation of the mitotic spindle at the onset of cytokinesis (*Sugioka and Bowerman, 2018*; *Matsumura et al., 2012*; *Kaltschmidt et al., 2000*; *Grill et al., 2001*; *Knoblich, 2010*) the mechanisms controlling reorientation remain not very well understood. In this work, we investigate cell

repositioning during cytokinesis in the early development of the *Caenorhabditis elegans* nematode. The fully grown hermaphrodite worm consists of exactly 959 somatic cells that are essentially invariant both in terms of position and lineage (*Sulston and Horvitz, 1977*; *Sulston et al., 1983*; *Schnabel et al., 2006*; *Li et al., 2019*). Development is deterministic from the start: the one-cell embryo undergoes an asymmetric cell division that gives rise to the AB (somatic) lineage and the P lineage (*Sulston et al., 1983*; *Bruce et al., 2002*). While the anterior daughter cell, AB, undergoes a symmetric cell division into ABa and ABp, the posterior daughter cell, $P_1$, divides asymmetrically into EMS forming the endoderm and mesoderm, and $P_2$ forming the germ line (*Sulston et al., 1983*). Appropriate cell-cell contacts are instrumental for development as they can determine cell identity (*Priess, 2005*; *Artavanis-Tsakonas et al., 1999*; *Mango et al., 1994*; *Mello et al., 1994*; *Moskowitz et al., 1994*). For example, reorientation of the ABa and ABp cells via pushing with a micro needle leads to an altered cell-cell contact pattern and an altered body plan with an inverted L/R body axis (*Wood, 1991*). Consequently, proper cell positioning, perhaps mediated via repositioning of the mitotic spindle during cytokinesis, is crucial (*Hennig et al., 1992*). Here, we set out to investigate which of the cells of the early embryo undergo reorientations during cytokinesis, and by which mechanism they do so.

Recently, a role for the actomyosin cell cortex in determining the cell division axis of early *C. elegans* blastomeres was identified (*Naganathan et al., 2014*; *Sugioka and Bowerman, 2018*). The actomyosin cortex is a thin layer below the plasma membrane that consists mainly of actin filaments, actin binding proteins and myosin motor proteins (*Pollard and Cooper, 1986*). Collectively, these molecules generate contractile forces that can shape the cell, drive cortical flows during polarization and orchestrate other active processes such as cell division (*Mayer et al., 2010*; *Pollard, 2017*). Cell-cell contacts can impact on the activity of myosin and the generation of contractile stresses, and the resultant pattern of cortical flows can determine the orientation of the mitotic spindle at the onset of cytokinesis (*Sugioka and Bowerman, 2018*). From a physical point of view, the actomyosin cortex can be thought of as a thin layer of a mechanically active fluid (*Jülicher et al., 2007*; *Ramaswamy and Simha, 2006*; *Salbreux et al., 2009*; *Mayer et al., 2010*) with myosin-driven active stress gradients generating cortical flows (*Mayer et al., 2010*). Interestingly, actomyosin can also exhibit rotatory flows driven by active torque generation. These chiral rotatory cortical flows reorient the ABa cell and the ABp cell during cytokinesis, driving a cell skew of ~20° during division (*Naganathan et al., 2014*). This skew results in a L/R asymmetric cell-cell contact pattern (*Pohl and Bao, 2010*), thus executing left-right (L/R) symmetry breaking in the entire organism. However, how general such reorientation events are, and how they are controlled, remains unclear. Furthermore, it remains poorly understood whether chiral flows are prevalent in other cell divisions as well, and if they play a prominent role in cell repositioning during early embryogenesis of the *C. elegans* nematode.

## Results

### Early cell divisions of the AB lineage, but not of the P/EMS lineage, undergo chiral counter-rotating actomyosin flows

We set out to quantify chiral rotatory flows in the actomyosin cell cortex of the first nine cell divisions in early *C. elegans* development. In order to image *C. elegans* embryogenesis, various mounting techniques have been described that either mildly compress the embryo or mount the embryo uncompressed. We first compared the degree of embryo compression using two common mounting methods (*Figure 1—figure supplement 1*): (1) Attaching the embryos to an agarose pad (*Bargmann and Avery, 1995*) and (2) embedding the embryos in low-melt agarose (*Naganathan et al., 2014*). As reported before (*Walston and Hardin, 2010*), we found that the first method indeed compressed the embryos and leads to an aspect ratio of 1.29 ± 0.14. Embedding embryos in low-melt agarose resulted in an aspect ratio of 1.02 ± 0.3 and hence a reduced compression. Measurements of early embryos in the adult uterus revealed an aspect ratio of 1.16 ± 0.7. (*Figure 1—figure supplement 1*), indicating that *in-utero* embryo compression strength is intermediate. Throughout this study, we used the classic mounting method using an agarose pad, unless stated otherwise.

We started with investigating the first two divisions, the divisions of the $P_0$ zygote and the AB cell, and quantified flows from embryos containing endogenously tagged non-muscle myosin-II (NMY-2::GFP) using spinning disc microscopy (*Videos 1* and *2*). We used particle image velocimetry (PIV) to determine cortical flow velocities in two rectangular regions of interest (ROIs) placed on opposite sides of the cytokinetic ring (*Figure 1A*, *Figure 1—figure supplement 2*). In particular, we were interested in chiral cortical flows that yield velocity contributions parallel to the cytokinetic ring, referred to as 'y-direction flows' (*Figure 1—figure supplement 2*). Beginning at the onset of furrow ingression, y-direction flows were averaged in each ROI over a time period of 21 s. This narrow time window provides a characterization of cortical flows restricted to mid-cytokinesis, and it ensures that shape changes of the dividing cell are minor and do not affect the velocity measurements. Note that, over the full time course from early to late cytokinesis, cortical flow velocities vary (*Figure 1—figure supplement 2*), while we did not observe major differences in the myosin distribution (*Figure 1—figure supplement 3*).

Strikingly, while y-direction flows in both ROIs of the $P_0$ cell occur with the same orientation (*Figure 1A*, left), they have opposite orientations in AB cells (*Figure 1A*, right), such that the two dividing cell halves of the AB cell effectively spin in opposite directions. We refer to such flows as chiral counter-rotating flows (*Figure 1—figure supplement 2*, *Videos 1* and *2*). To quantify the speed and the handedness of counter-rotating flows more generally, we use the measured velocities in each ROI, denoted as $\underline{v}_1$ and $\underline{v}_2$, and define a chiral counter-rotation flow velocity $v_c = \underline{e}_z \cdot (\underline{e}_x \times \underline{v}_2 - \underline{e}_x \times \underline{v}_1)$, where $\underline{e}_x$ is a unit length base vector pointing from the cytokinetic ring toward the pole, $\underline{e}_y$ is an orthogonal unit length vector parallel to the cytokinetic plane (*Figure 1A*), and $\underline{e}_z = \underline{e}_x \times \underline{e}_y$. With this definition, $|v_c|$ quantifies the speed of counter-rotating flows and the sign of $v_c$ denotes their handedness (see Materials and methods and *Figure 1—figure supplement 2*). For the AB cell division, we then find $v_c = -7.01 \pm 0.34$ µm/min (mean ± error of the mean at 95% confidence, unless otherwise stated), indeed indicating the presence of counter-rotating flows with consistent handedness. Furthermore, for the division of the $P_0$ cell, we find that counter-rotating flows are essentially absent ($v_c = 0.01 \pm 0.51$ µm/min). Instead, the cortical flows of dividing $P_0$ cells, where both cell halves move in the same direction, correspond to net-rotating flows that resemble whole embryo rotations (*Schonegg et al., 2014*). We conclude that out of the first two cell divisions in the developing nematode, only one is accompanied by chiral counter-rotating actomyosin flows.

This raises the question which of the later cell divisions display chiral counter-rotating flows, and which do not. To answer this, we quantified chiral counter-rotating flows for the next seven cell divisions as described above. We found that cells of the AB lineage (ABa, ABp, ABar and ABpr) exhibit chiral counter-rotating flows with $v_c$ = $-3.46 \pm 0.33$ µm/min, $-3.68 \pm 0.41$ µm/min, $-1.24 \pm 0.19$ µm/min and $-1.16 \pm 0.22$ µm/min respectively (*Figure 1B*). In contrast, average chiral counter-rotating flow velocities of cells of the P/EMS lineage ($P_1$, EMS and $P_2$) are small: 0.15 ±0.38 µm/min, 0.03 ± 0.14 µm/min and 0.25 ±

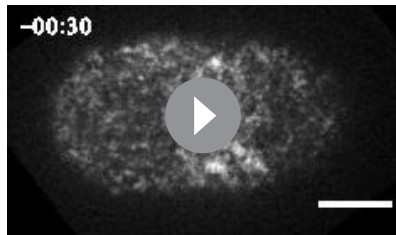

**Video 1.** Movie of the actomyosin cortex during the first cell division. The cortex is marked by NMY-2::GFP. The $P_0$ cell does not exhibit chiral counter-rotating flows during its division. Cortical flows in the two dividing halves of the $P_0$ cell flow in the same y-direction. Scale bar 10 µm.

https://elifesciences.org/articles/54930#video1

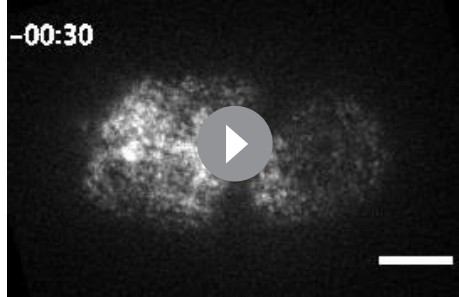

**Video 2.** Movie of the actomyosin cortex during the second cell division. The AB cell division exhibits chiral counter-rotating flows. The two dividing halves of the AB cell counter rotate in opposite y-directions. Scale bar 10 µm.

https://elifesciences.org/articles/54930#video2

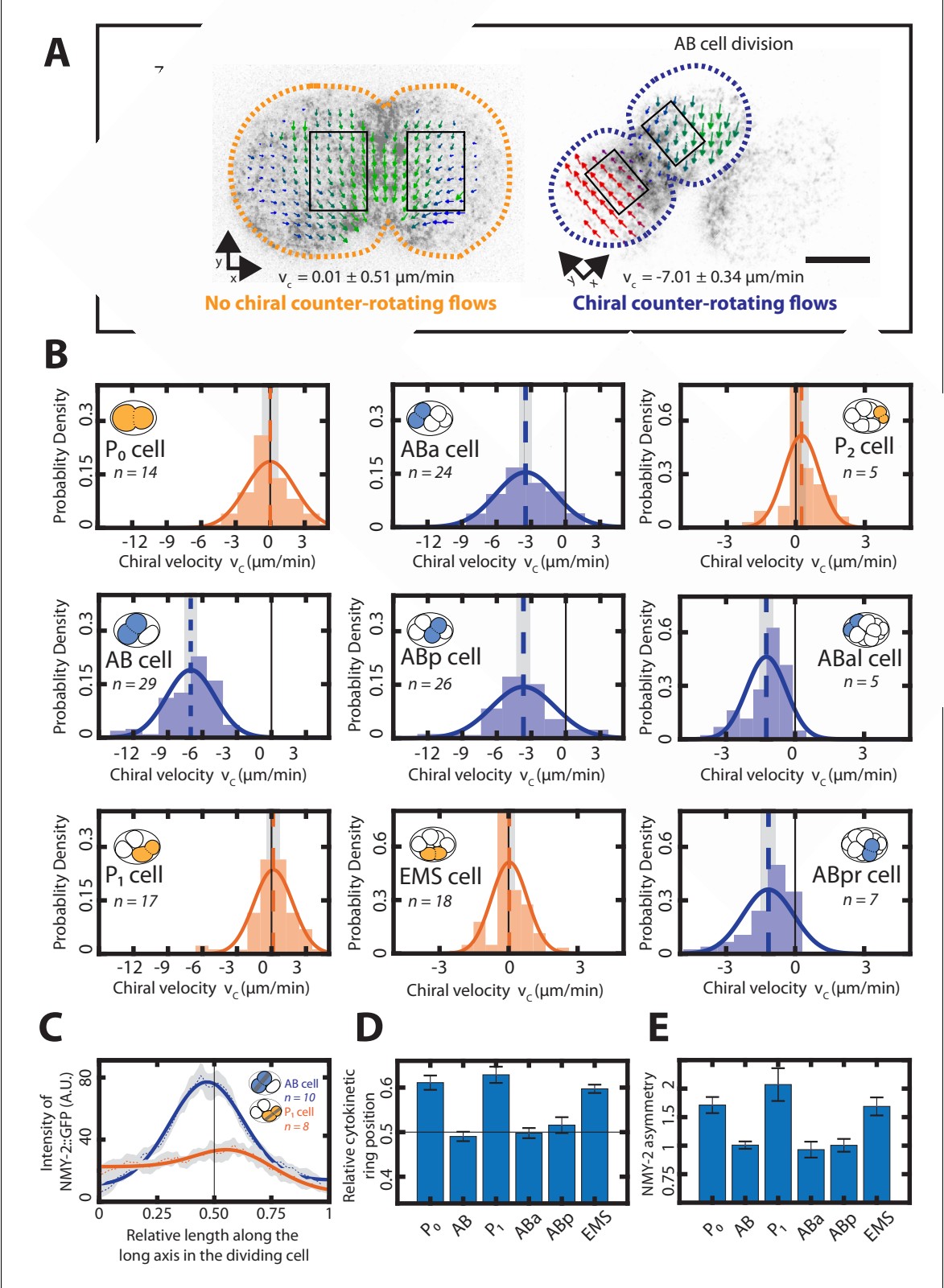

**Figure 1.** Chiral counter-rotating flows are cell-lineage specific. (**A**) Representative images of cortical myosin (grey, NMY-2::GFP) for the first ($P_0$, yellow) and the second (AB, blue) cell division of *C. elegans* embryos. Arrows indicate the cortical flow field as measured by PIV and time-averaged over 21 s and over the onset of cytokinesis. Arrow colors indicate the y-direction velocity (parallel to the cytokinesis furrow, coordinate systems are indicated). Dotted lines indicate cell boundaries, black boxes represent the regions of interest used for calculating velocities. Scale bar, 10 µm. (**B**) Histograms of

*Figure 1 continued on next page*

*Figure 1 continued*

the instantaneous chiral counter-rotating flow velocity $v_c$ (see Materials and methods for definition). Solid lines indicate the best-fit gaussian probability density function. Dotted vertical colored lines represent the mean $v_c$, grey boxes represent the error of the mean. Thin black solid lines indicate a chiral flow velocity of zero. Inset, colored cells indicate the cell analyzed; AB lineage in blue, P/EMS lineage in yellow. (C) Dotted lines with shaded error region indicate the average myosin concentration profile along the long axis of the AB (blue) and $P_1$ (orange) dividing cell, solid lines represent a best fit with a combined step and gaussian function (see Materials and methods) Inset, colored cells indicate the cell analyzed; grey stripe indicates the region used for averaging. (D) Average relative position of the cytokinetic ring along the long axis of the dividing cell for the first 6 cell divisions. Black thin lines in (C) and (D) indicate the center of the cell (n = 8 for all cell divisions). (E) Myosin asymmetry ratio (Anterior [NMY-2::GFP]/Posterior [NMY-2::GFP]) (see Materials and methods) for the first six-cell divisions along the long axis of the dividing cell (n = 8 for all cell divisions). Errors indicate the error of the mean at 95% confidence.

The online version of this article includes the following figure supplement(s) for figure 1:

**Figure supplement 1.** Effect of embryo compression on embryo thickness in AP-DV and LR-DV plane.

**Figure supplement 2.** Time evolution of chiral counter-rotating velocity during cytokinesis.

**Figure supplement 3.** Evolution of myosin distibution profile along the cell division axis of the AB cell.

**Figure supplement 4.** Early cell divisions display different types of rotatory flows.

**Figure supplement 5.** Myosin density and total flow velocity in development.

**Figure supplement 6.** Determining myosin ratio during cell division.

**Figure supplement 7.** Active chiral fluid theory can recapitulate the flows observed during early development.

**Figure supplement 8.** Thin film active chiral fluid theory (see Appendix) can recapitulate the contractile and rotatory flows observed in the cells of the AB-lineage.

**Figure supplement 9.** Chiral counter-rotating flows during first cell division in *par-2 (RNAi)* embryos.

**Figure supplement 10.** Symmetric cell division is required for chiral flows to emerge.

0.19 µm/min, respectively (*Figure 1B*). Like $P_0$, cells of the P/EMS lineage, instead exhibit net-rotating flows, where the y-direction flows have the same orientation in both cell halves (*Figure 1—figure supplement 4*, *Videos 1–5*). To conclude, early cell divisions of the AB lineage, but not of the P/EMS lineage, undergo chiral counter-rotating flows. We note that due to a decrease in the overall cortical flow speed, also the chiral counter-rotating flow velocity $v_c$ in the AB lineage decreases as development progresses and cells become smaller (*Figure 2B*, *Figure 1—figure supplement 5*). This is accompanied by a decrease in the myosin concentration in the cytokinetic ring (*Figure 1—figure supplement 5*) and is consistent with overall contractility being reduced when development progresses. Together, these findings show that both the presence and the strength of chiral counter-rotating flows are related to the fate of the dividing cell.

We have shown previously that chiral counter-rotating cortical flows are driven by active torque generation that depends on the myosin distribution (*Naganathan et al., 2014*). Given that chiral counter-rotating flows arise only during AB and not P/EMS cell divisions, we hypothesize that myosin distributions during division of the AB and the P/EMS-lineage cells differ in specific features. To investigate this, we determined the NMY-2 distribution along the long axis of dividing cells for the first six cell divisions (*Figure 1C*), and extracted two features (see Materials and methods): (1) a myosin ratio that characterizes the difference in myosin intensity between the two cell halves (*Figure 1D*, *Figure 1—figure supplement 6*) and (2) the relative myosin peak position along the long axis, which characterizes the cytokinetic ring position and the degree of asymmetry in the cell division (*Figure 1E*). We note that these quantities could be coupled, either mechanically (*Sedzinski et al., 2011*) or via par polarity complexes that facilitate cellular myosin asymmetry (*Munro et al., 2004*; *Gross et al., 2019*) and asymmetric positioning of the cleavage plane (*Galli and van den Heuvel, 2008*; *Colombo et al., 2003*). We find that myosin ratios in dividing AB, ABa and ABp cells are 1.0 ± 0.13, 0.92 ± 0.29, and 1.00 ± 0.24, respectively, indicating that there is no significant difference in cortical myosin concentration between the two halves of dividing cells from the AP lineage. In contrast, the myosin ratios in $P_0$, $P_1$ and EMS are 1.67 ± 0.3, 2.07 ± 0.61, 1.68 ± 0.33, respectively (*Figure 1E*). Furthermore, we find that the relative position of myosin peaks along the long axis of dividing AB, ABa and ABp cells are 0.49 ± 0.01, 0.49 ± 0.01 and 0.51 ± 0.02, respectively. In contrast, myosin peaks in dividing $P_0$, $P_1$ and EMS cells are positioned at 0.58 ± 0.01, 0.57 ± 0.02 and 0.59 ± 0.01 relative to cell length, respectively (*Figure 1D*). To conclude, the early cell divisions of the AB lineage are symmetric in terms of myosin intensity ratio and cytokinetic ring position, while early cell divisions of the P/EMS lineage are asymmetric in these two features.

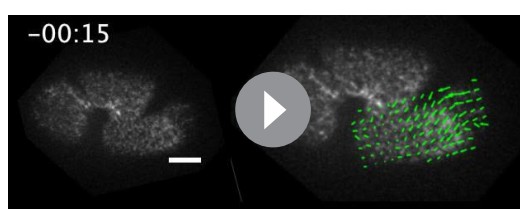

**Video 3.** Movie of the actomyosin cortex during the third cell division (left) and with PIV flow fields (right). The cortex is marked by NMY-2::GFP. The $P_1$ cell division does not exhibit chiral counter-rotating flows. The nature of the flow is similar to the one observed in case of $P_0$ cell division. Scale bar 10 μm.
https://elifesciences.org/articles/54930#video3

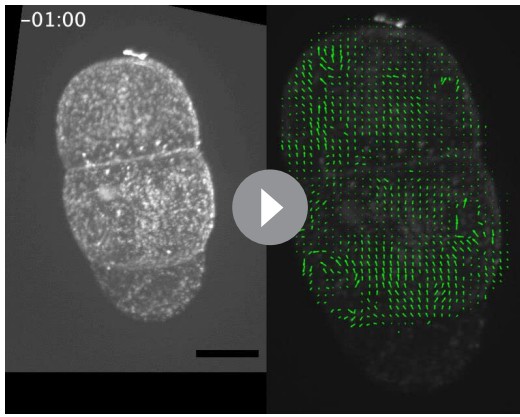

**Video 4.** Movie of the actomyosin cortex during the ABa and the ABp cell divisions (left) and with PIV flowfield (right). The cortex is marked by NMY-2::GFP. ABa and ABp exhibit chiral counter-rotating flows during division similar to the flows observed in the AB cell division. Scale bar 10 μm.
https://elifesciences.org/articles/54930#video4

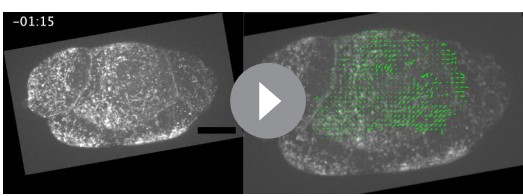

**Video 5.** Movie of the actomyosin cortex during the EMS division (left) and with PIV flowfield (right). The cortex is marked by NMY-2::GFP. No chiral counter-rotating flows are observed during the EMS division. Scale bar 10 μm.
https://elifesciences.org/articles/54930#video5

We next asked if the presence and absence of chiral counter-rotating flows in AB and P/EMS cells can be accounted for by the observed difference in myosin ratio and cytokinetic ring position. To this end, we utilized a thin film active chiral fluid theory (*Fürthauer et al., 2013*; *Naganathan et al., 2014*) that describes active chiral processes in a fluid-like material and the flows these processes are expected to give rise to. We then considered a minimal model that allows to compute $y$-direction flows, and hence chiral counter-rotating flow velocities $v_c$, from a given myosin ratio and cytokinetic ring position (Appendix, *Videos 6–8*). We find that a symmetric myosin distribution and cytokinetic ring position (myosin ratio $\approx 1$, relative myosin peak position $\approx 0.5$) result in a maximum value of $v_c$. Accordingly, myosin ratios different from one and an asymmetric ring position result in smaller values of $v_c$, where the model suggests that $v_c$ is more sensitive to changes in the myosin ratio than to changes in the ring position (*Figure 1—figure supplement 7*).

Our results indicate that the switch between chiral and non-chiral flows between the two lineages could in principle be attributed to the observed difference in myosin ratio, cytokinetic ring position or both (*Appendix 1—table 1*). We note, however, that while the thin film active chiral fluid theory can quantitatively relate y-direction flows of dividing AB, ABa, and ABp cells with the myosin profiles along their long axis (*Figure 1—figure supplement 8*, *Appendix 1—table 1*), this is not possible for the P/EMS lineage. We suspect that this is due to specific features of the overall dynamics that are not included in our theoretical description, such as movements of the midbody remnant particular to the P/EMS linage (*Singh and Pohl, 2014*), as well as more general aspects that we have neglected for simplicity, such as the shape dynamics of the dividing cell (*Mietke et al., 2019*) and possible inhomogeneities in friction forces due to mechanical interactions with the surrounding. This indicates that our understanding of the mechanisms that lead to the cortical flows in P/EMS linage divisions is still incomplete. Taken together, our analysis implies a mechanism whereby the occurrence of chiral counter-rotating actomyosin flows depends on the amount of cortical myosin in the cell halves and the position of the cytokinetic ring.

Next, we asked whether symmetric cell division is indeed required for chiral counter-rotating flows to emerge. The first cell division in wild-type embryos is asymmetric and does not display chiral counter-rotating flows. The asymmetry of the $P_0$ division is controlled by Par polarity complexes (*Kemphues et al., 1988*) and depletion of the posterior Par protein, PAR-2, led to a symmetric myosin distribution with a myosin ratio of one and

centered ring position (*Figure 1—figure supplement 9*). Strikingly, *par-2* depletion also triggered the emergence of chiral counter-rotating flow during $P_0$ cytokinesis ($v_c = -5.71 \pm 0.89$ µm/min; *Figure 1—figure supplement 9*) demonstrating that inducing symmetric cell division via *par-2 (RNAi)* is sufficient to trigger chiral counter-rotating flows in $P_0$. Moreover, our hydrodynamic theory predicts that a centered ring position should result in chiral counter-rotating flows regardless of myosin asymmetry (*Figure 1—figure supplement 7*). To test this, we performed *lin-5 (RNAi)* treatment and analyzed cortical actomyosin flows. LIN-5 is involved in generating asymmetric pulling forces that displace the spindle from the center (*Galli and van den Heuvel, 2008*; *Lorson et al., 2000*). We first confirmed that indeed the myosin distribution in $P_0$ is still asymmetric in these embryos (albeit reduced when compared with control), while the ring position is at the center (*Figure 1—figure supplement 10*). Quantifying cortical actomyosin flows revealed that *lin-5 (RNAi)* triggered the emergence of chiral counter-rotating flows ($v_c = -1.73 \pm 0.74$ µm/min; *Figure 1—figure supplement 10*). Given that cortical myosin levels upon *lin-5 (RNAi)* are comparable to those observed in control, the emergence of chiral counter-rotating flows is unlikely to be the result of higher overall actomyosin activity (*Figure 1—figure supplement 10*). Although we cannot fully uncouple the effect of myosin asymmetry and ring position, these experimental results are qualitatively consistent with our theoretical predictions and thus suggest that both, asymmetries in myosin distribution and ring position, counteract the emergence of chiral counter-rotating flows. We conclude that symmetric cell division is required for chiral counter-rotating actomyosin flows to emerge during cytokinesis.

## Cells exhibiting chiral counter-rotating actomyosin flows also undergo spindle skews

We next focus our attention on the mechanism by which dividing cells assume their correct position inside the developing embryo. Specific rotations of the spindle (termed spindle skews) are thought to be a key mechanism by which dividing cells reposition themselves in the embryo (*Naganathan et al., 2014*; *Singh and Pohl, 2014*; *Liro and Rose, 2016*; *Bergmann et al., 2003*). One hypothesis is that spindle skews are driven by spindle elongation (*Hennig et al., 1992*). In this scenario, spindle elongation causes a skew because the spindle rearranges within an asymmetric cell shape, thus responding to the constraints provided by neighboring cells and the eggshell (*Figure 3A*). An alternative hypothesis is that spindle skews are driven by chiral counter-rotating flows of the dividing cell halves (*Naganathan et al., 2014*). In this case, the mechanism is akin to a bulldozer rotating on the spot by spinning its chains in opposite directions. If the cortical surface experiences a spatially inhomogeneous friction, e.g. when the friction with the eggshell is different

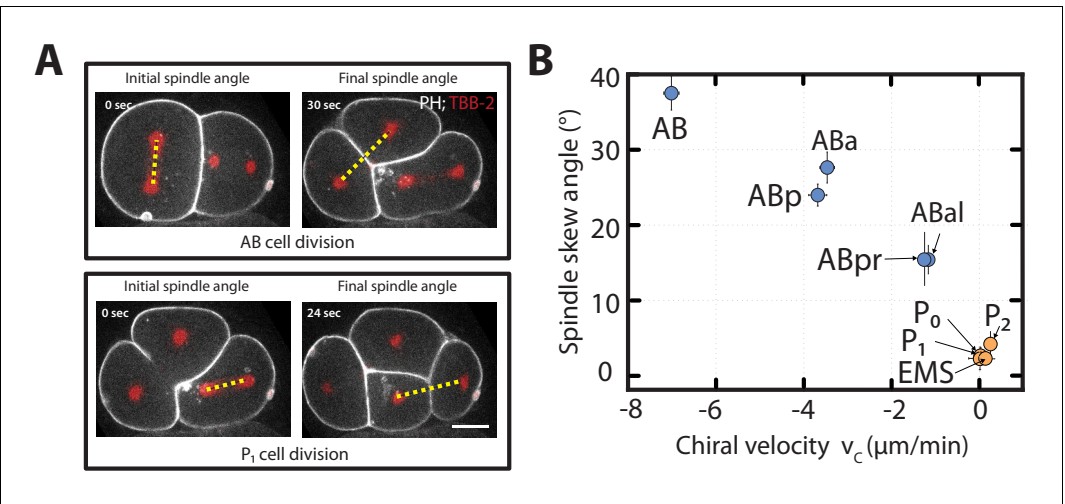

**Figure 2.** Chiral counter-rotating flow velocity and spindle skew decreases as development progresses. (**A**) Representative images show the angle of the mitotic spindle (yellow dotted line) before the onset (left) and at the end (right) of the second (AB, top) and the third ($P_1$, bottom) cell division. White, cell membrane imaged by PH:: GFP; red, spindle poles as imaged by TBB-2::mCherry. (**B**) Spindle skew angle (final minus initial spindle angle) vs. chiral velocity for the first nine cell divisions (N = 8 for all cell diviisons).

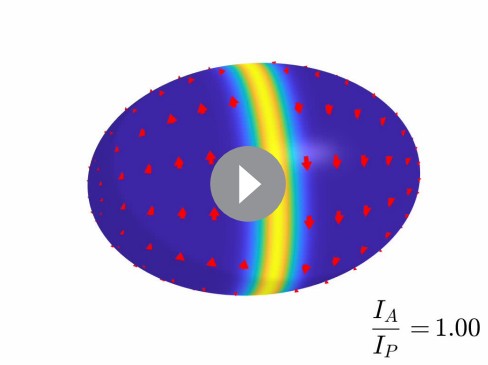

$$\frac{I_A}{I_P} = 1.00$$

**Video 6.** Chiral surface flows on an ellipsoidal surface for a centered contractile ring, $x_r/L = 0.5$, and varying myosin asymmetries.
https://elifesciences.org/articles/54930#video6

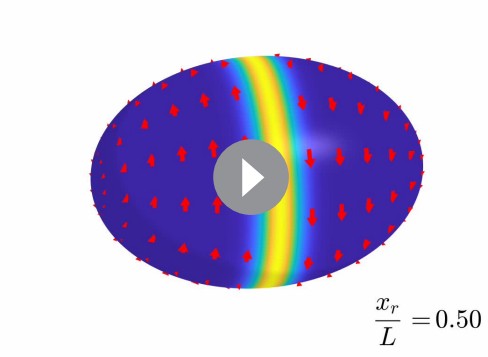

$$\frac{x_r}{L} = 0.50$$

**Video 7.** Chiral surface flows on an ellipsoidal surface for a symmetric myosin profile, $I_A/I_P = 1$ and varying positions of the contractile ring.
https://elifesciences.org/articles/54930#video7

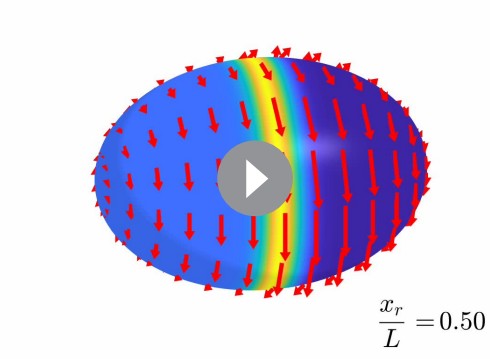

$$\frac{x_r}{L} = 0.50$$

**Video 8.** Chiral surface flows on an ellipsoidal surface for an asymmetric myosin profile, $I_A/I_P = 1.5$ and varying positions of the contractile ring.
https://elifesciences.org/articles/54930#video8

from the friction with a neighboring cell, chiral counter-rotating cortical flows of the two cell halves will lead to a torque that acts on the cell and can lead to a cell skew (*Figure 3A*). In this scenario, the cell orientation changes over time, and the spindle simply aligns accordingly.

To discriminate between these two hypotheses, we first asked if those cells that undergo spindle skews also exhibit chiral counter-rotating flows. We quantified spindle skews by determining the positions of spindle poles at the beginning and at the end of cytokinesis in embryos expressing TBB-2::mCherry for the first 11 cell divisions (*Video 9*). The angle between the lines that join the two spindle poles at the onset and after cell division defines the spindle skew angle (*Figure 2A*). We find that cells of the early AB lineage undergo an average skew angle of 21.17 ± 3.22° (AB: 37.51 ± 2.51°, ABa: 27.64 ± 2.29°, ABp: 23.98 ± 1.19°, ABal: 17.73 ± 1.98°, ABar: 15.51 ± 3.51°, ABpl: 10.43 ± 1.92° and ABpr: 15.42 ± 2.11°). In comparison, P/EMS lineage cells undergo significantly reduced spindle skews with an average skew angle of of 2.32 ± 0.43° ($P_0$: 2.31 ± 1.67°, $P_1$: 2.25 ± 0.66°, EMS: 2.66 ± 1.25°, $P_2$: 4.36 ± 1.67°). Note also, that spindle skew angles in the AB lineage decrease as development progresses, while the P/EMS lineage spindle skew angles remain small and approximately constant (*Figure 2B*). Given that AB lineage cells, but not P/EMS lineage cells, undergo chiral counter-rotating flows, we conclude that cells that exhibit significant chiral counter-rotating flows also undergo significant spindle skews during early development (*Figure 2B*).

In our earlier work, we showed that the skew of the ABa and ABp cells is in fact a three-dimensional process, tilting the spindle in both the anteroposterior-left-right (AP-LR) plane and in the orthogonal left-right-dorsoventral (LR-DV) plane (*Naganathan et al., 2014*). Therefore, in addition to the well known skew of the AB cell in the anteroposterior-dorsoventral (AP-DV) plane that sets up the dorsoventral axis, we hypothesized that there might be an additional skew in the orthogonal LR-DV plane (*Figure 3E*, *Figure 3—figure supplement 1*). In order to test this hypothesis, we examined spindle orientation throughout AB cell cytokinesis from an anterior view down the anteroposterior axis, but did initially not observe any obvious skews in the orthogonal LR-DV plane (*Figure 3—figure supplement 1*). However, as discussed above, the embryo mounting method used in this study slightly compresses the embryos along their left-right axis, which could constrain spindle movements along this direction. In order

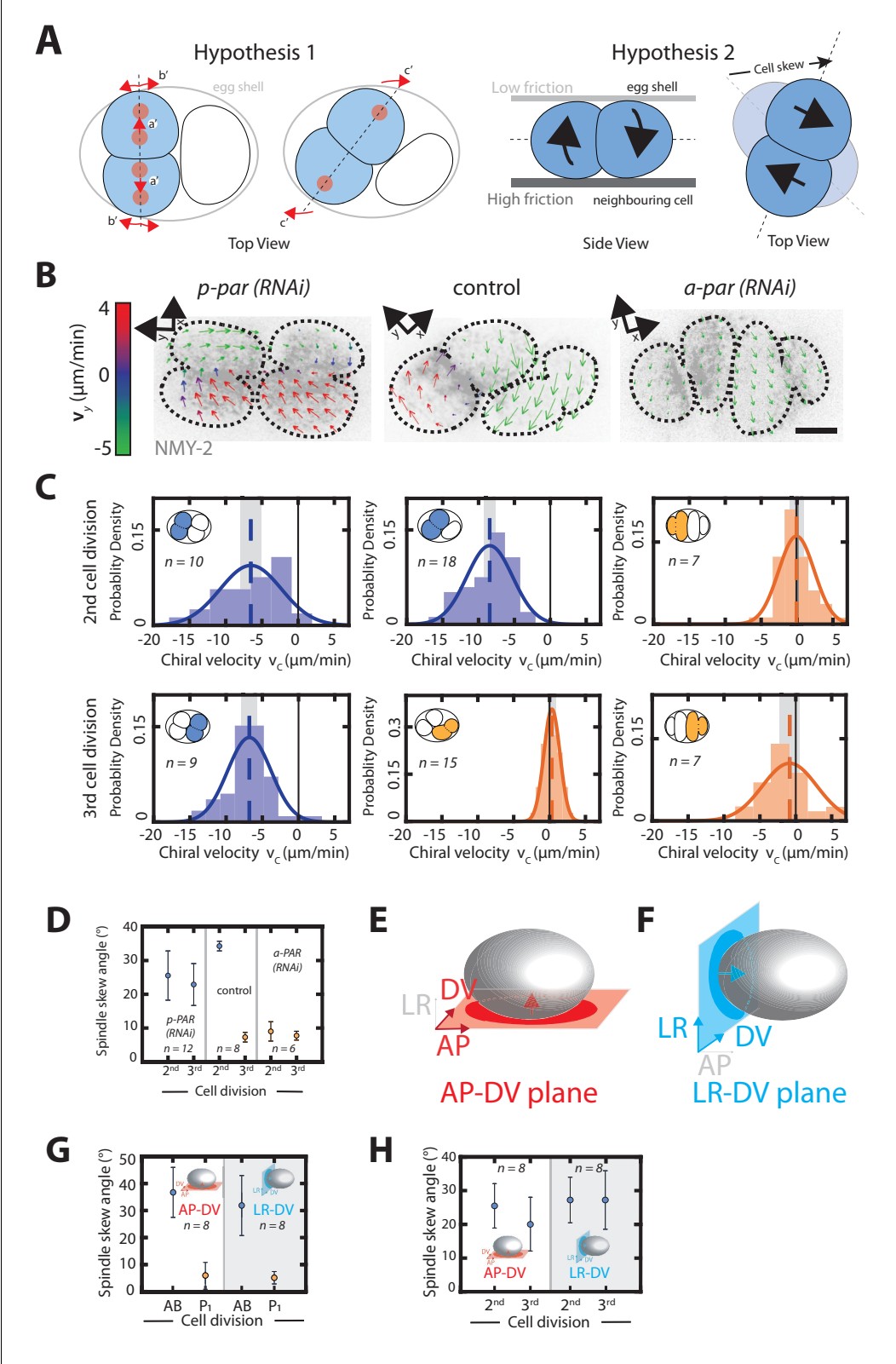

**Figure 3.** Chiral counter-rotating flows are cell-lineage dependent. (**A**) Possible mechanisms of spindle skew and cell rearrangement. Hypothesis one: spindle skews are driven by spindle elongation. The mitotic spindle initially elongates along the dorsoventral axis (a' and b'), but then skews (c') because there is not enough space given the physical constraints imposed by the eggshell. Dashed line depicts the cell division axis, red dots depict spindle poles, arrows indicate spindle pole movements. Hypothesis two: spindle skews are driven by counter-rotating flows of the dividing cell halves.

*Figure 3 continued*

Left: top view, right: side view. Chiral counter-rotation of the two halves of a dividing cell gives rise to a cell skew if the dividing cell experiences non-uniform friction with its surroundings, e.g. a lower friction with the eggshell above and a higher friction with the cell below. (**B**) Representative images of cortical myosin (grey, NMY-2::GFP) for the second and the third cell division of *C. elegans* embryos. Arrows indicate the cortical flow field as measured by PIV and time-averaged over 21 s and over the onset of cytokinesis. Arrow colors indicate the y-direction velocity (parallel to the cytokinesis furrow, coordinate systems are indicated). (**C**) Histograms of the instantaneous chiral counter-rotating flow velocity $v_c$ for *par* perturbations (see Materials and methods for definition). Solid lines indicate the best-fit gaussian probability density function. Dotted vertical colored lines represent the mean $v_c$, grey boxes represent the error of the mean. Thin black solid lines indicate a chiral flow velocity of zero. Inset, colored cells indicate the cell analyzed; AB lineage in blue, P/EMS lineage in orange. (**D**) Spindle skew angle for control (L4440), *par-2; chin-1; lgl-1 (RNAi)* and *par-6 (RNAi)* embryos. (**E, F**) Schematic of a *C. elegans* embryo. 3-D recordings capturing spindle pole movements and corresponding spindle skews were projected on either the (**E**) anteroposterior-dorsoventral (AP-DV) plane (red) or the (**F**) left-right-dorsoventral (LR-DV) plane (blue). (**G, H**) Spindle skew angles during the second and third cell division in control (**G**) and upon *par-2 (RNAi)* (**H**) projected onto the anteroposterior-dorsoventral plane (termed AP-DV plane, left) or onto the dorsoventral-left-right plane (termed LR-DV plane, right). The embryo schematic denotes the plane of projection (rectangle) and the direction of view (arrow) as described above. Scale bars, 10 μm. Errors indicate the error of the mean at 95% confidence.

The online version of this article includes the following figure supplement(s) for figure 3:

**Figure supplement 1.** 3-D spindle skew dynamics in the AB cell and the $P_1$ cell.
**Figure supplement 2.** 3-D Spindle skew dynamics in *par-2 (RNAi)* embryos.

---

to determine whether the spindle skew in the LR-DV plane is concealed by embryo compression, embryos were embedded in low-melt agarose to prevent compression and subsequently analyzed. Chiral counter-rotating flows in the AB cell could still be observed and the spindle skew in the AP-DV plane was similar in compressed and uncompressed conditions (37.51 ± 2.51° and 36.67 ± 9.24°, respectively). However, the absence of embryo compression revealed a significant clockwise skew in the LR-DV plane (when viewed from the anterior) with a skew angle of 31.85 ± 11.11° (***Figure 3G***, ***Figure 3—figure supplement 1***, ***Video 10***). We did not observe any significant skew in the $P_1$ cell in the LR-DV plane (***Figure 3—figure supplement 1***). These findings show that skews of the AB cell division axis are in fact also a three-dimensional process, where the skew occurs in both the AP-DV and LR-DV planes. Note that, under the assumption that the friction forces at the interface between AB and $P_1$ cell are high compared to those between AB cell and the surrounding vitelline layer and egg shell, the handedness of the chiral counter-rotating flows is consistent with the observed orientation of the spindle skew in the LR-DV plane. Using the same argument, the skew in the AP-DV plane could in principle result from friction forces between the AB cell and the egg shell that are different on the left and on the right side (see section 'Left-right asymmetric friction forces can account for the skew of the AB cell' for a further analysis of this hypothesis).

The results obtained so far are consistent with a scenario in which cell fate determines the presence of chiral counter-rotating flows, and these chiral counter-rotating flows then drive spindle skews. We test for these two causal relationships separately. We first asked if cell fate determines the presence of chiral counter-rotating flows. In this case, we expect that on the one hand, anteriorizing the worm leads to equal and significant

**Video 9.** Movie of maximum intensity projection of dividing embryo from one-cell stage to 24-cell stage. Spindle apparatus and spindle poles are marked by TBB-2::mCherry (red) and membrane (grey) is marked by PH::GFP. Scale bar 10 μm.
https://elifesciences.org/articles/54930#video9

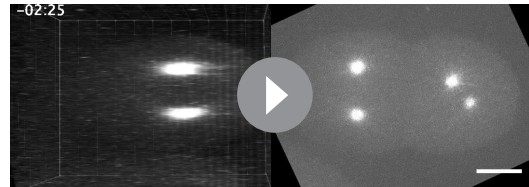

**Video 10.** Spindle skew in the AB cell imaged in uncompressed wild type embryos (SWG63) projected along the (left) LR-DV plane and (right) AP-DV plane by visualizing TBB-2::mCherry. Scale bar 10 μm.
https://elifesciences.org/articles/54930#video10

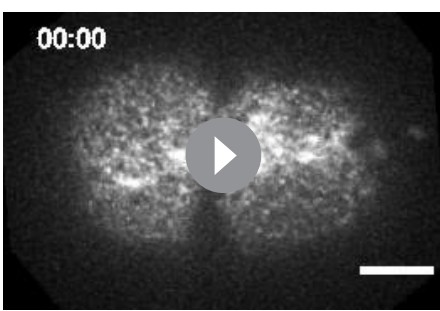

**Video 11.** Chiral counter-rotating cortical flows during the second and third cell division in *par-2; chin-1; lgl-1 (RNAi)* embryos. The cortex is marked by NMY-2::GFP. Scale bar 10 μm.

https://elifesciences.org/articles/54930#video11

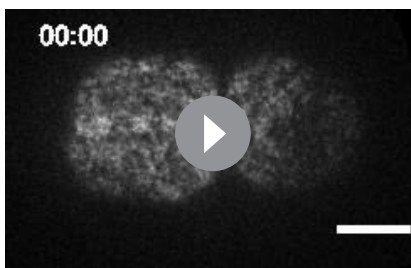

**Video 12.** Cortical flows during the second and third cell division in control embryos. The second cell division exhibits chiral counter-rotating flows whereas the third cell division shows absence of chiral counter-rotating flows. Scale bar 10 μm.

https://elifesciences.org/articles/54930#video12

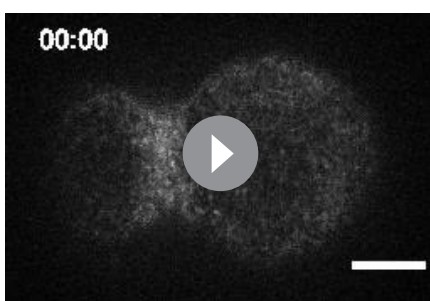

**Video 13.** Cortical flows during the second and third cell division in *par-6 (RNAi)* embryos. No chiral counter-rotating flows are observed during either of these cell divisions. Scale bar 10 μm.

https://elifesciences.org/articles/54930#video13

chiral counter-rotating flows for the second and third division. On the other hand, we expect that posteriorizing the worm would lead to an absence of chiral counter-rotating flows. Consistent with the first expectation, anteriorizing the embryo via *par-2; chin-1; lgl-1 (RNAi)* (*Beatty et al., 2013*) results in the second and the third cell division exhibiting chiral counter-rotating flows with $v_c$ values that are not significantly different from one another ($-6.53 \pm 1.13$ μm/min and $-6.67 \pm 0.8$ μm/min, respectively; Wilcoxon rank sum test at 95% confidence; *Figure 3B,C*). Consistent with the second expectation, posteriorizing the embryo via *par-6 (RNAi)* (*Hung and Kemphues, 1999*) results in both the second and the third cell division not exhibiting counter-rotating flows ($v_c = 0.07 \pm 0.71$ μm/min and $-0.85 \pm 1.13$ μm/min, respectively; *Figure 3B,C*, *Videos 11–13*). Hence, cell fate determines the presence or absence of chiral counter-rotating flows. Furthermore, switching the cell fate results in concomitant changes in the spindle skew angle in the AP-DV plane for both the second and third cell division in compressed as well as non-compressed embryos (*Figure 3D,H*). Strikingly, we observed that spindles of the two AB-like cells in uncompressed anteriorized embryos undergo clockwise skew in the LR-DV plane when viewed from the closest pole (skew angles of each AB-like cell = $26.93 \pm 6.78°$ and $27.44 \pm 8.66°$) (*Figure 3H*, *Figure 3— figure supplement 2*, *Videos 14–16*). Again, assuming that the friction forces at the cell-cell contact surface are high, the orientation of these skews is consistent with the handedness of the observed chiral counter-rotating flows (*Figure 3— figure supplement 1*). We conclude that cell fate determines both the presence of chiral counter-rotating flows and the degree as well as the direction of spindle skew.

## Chiral counter-rotating flows drive spindle skews

We next tested the second causal relationship, and asked if chiral counter-rotating flows in the cortex drive spindle skews. Borrowing from the analogy of a bulldozer, for which the speed of the chiral counter-rotating chains sets the speed at which the entire bulldozer rotates, we investigated if increasing or decreasing the chiral counter-rotating flow velocities results in corresponding changes of the spindle skew rate. We first evaluated if we can increase and decrease chiral counter-rotating flow velocities in the AB cell, by RNAi of RhoA regulators. Weak perturbation RNAi of *ect-2* results in $v_c = -3.57 \pm 0.57$ μm/min, which is reduced in comparison to the unperturbed embryo ($v_c = -6.6 \pm 0.37$ μm/min, Wilcoxon rank sum test at 95% confidence, *Figure 4A,B*). Conversely, weak perturbation RNAi of *rga-3* increases the chiral counter-rotating flow velocity to $v_c = -8.71 \pm 0.58$ μm/min (*Figure 4A,B*). Note that these perturbations impact only chiral counter-rotating flows, and do not significantly change

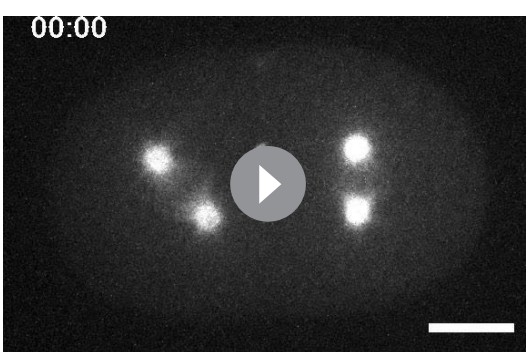

**Video 14.** Spindle skew in uncompressed *par-2 (RNAi)* anteriorized embryos during third and fourth cell division projected in the AP-DV plane by visualizing TBB-2::mCherry. Embryos were mounted using low melt agarose method. Scale bar 10 µm.
https://elifesciences.org/articles/54930#video14

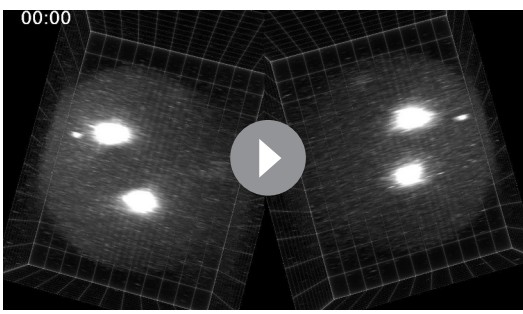

**Video 15.** Spindle skew in uncompressed *par-2 (RNAi)* anteriorized embryos during (left) third and (right) fourth cell division projected along the (left) LR-DV plane viewed from anterior end and (right) LR-DV plane viewed from posterior end by visualizing TBB-2::mCherry. Scale bar 10 µm.
https://elifesciences.org/articles/54930#video15

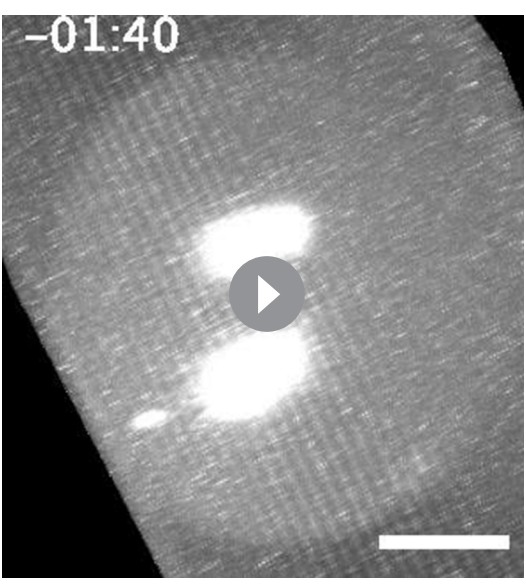

**Video 16.** Spindle skew in uncompressed *par-2 (RNAi)* anteriorized embryos in two juxtaposed AB-like cells when projected along LR-DV plane and viewed from anterior end. Spindle poles are visualized using TBB-2::mCherry. Scale bar 10 µm.
https://elifesciences.org/articles/54930#video16

overall contractility, as illustrated by the on-axis flow into the contractile ring, which is not significantly altered (Wilcoxon rank sum test at 95% confidence, *Figure 4C*). We now use these perturbations to test whether chiral counter-rotating flows determine the rate of spindle skew. We indeed find that decreasing the chiral counter-rotating flow velocity by mild *ect-2 (RNAi)* leads to concomitant reduction of the rate of spindle skew (average peak rate: 0.59 ± 0.05°/min, *Video 17*) as compared to 1.01 ± 0.1°/min in control embryos (*Figure 4D*, *Video 18*). Conversely, increasing RhoA signaling by mild *rga-3 (RNAi)* treatment leads to an increase of both the chiral counter-rotating flow velocity and the rate of spindle skew (average peak rate: 1.29 ± 0.12°/min (*Video 19*, *Figure 4D*). Note that the peak rates of spindle elongation remain unchanged as compared to the control for both perturbations (*Figure 4D*, *Figure 4—figure supplement 1*). We conclude that changing chiral counter-rotating flow velocities results in concomitant changes of the rates of spindle skews without impacting on the dynamics of spindle elongation. In addition, given that the contractility-driven flows remain unaltered upon *rga-3 (RNAi)* and *ect-2 (RNAi)*, overall actomyosin contractility is unlikely to contribute significantly to cell division skews. These results instead support our second hypothesis, that chiral counter-rotating flows mechanically drive cell skews and consequently also spindle skews.

The position of the mitotic spindle is determined by an evolutionary conserved protein complex, containing GPR-1/2 and its interacting partner LIN-5/NuMa, that facilitates pulling forces on astral microtubules at the cell cortex (*Srinivasan et al., 2003*; *Galli et al., 2011*; *Colombo et al., 2003*). Therefore, we next asked whether spindle skews during anaphase of dividing AB lineage cells are dependent on this known spindle orientation pathway. To address this, we analyzed actomyosin flows and the cell division skew of AB upon depletion of *lin-5* and *gpr-1/2*. Both perturbations resulted in defects that have been previously described, including symmetric cell division of $P_0$ (*Figure 1—figure supplement 10* and data not shown) and misaligned spindles in the $P_1$ and AB cells

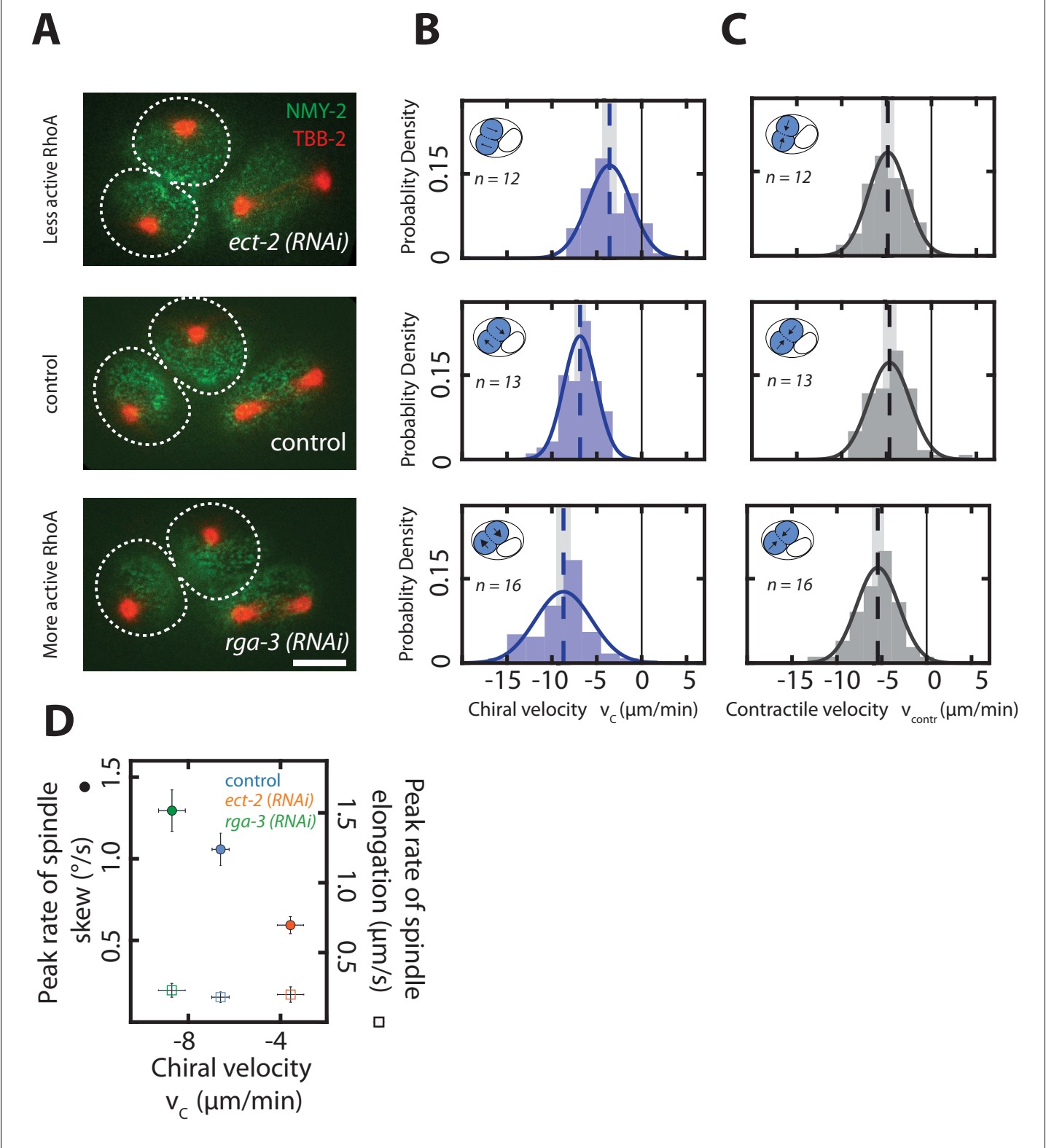

**Figure 4.** Chiral counter-rotating flow velocity determines the rate of spindle skew in the AB cell. (**A**) Fluorescence images of two-cell embryos during cytokinesis upon *ect-2 (RNAi)* and *rga-3 (RNAi)*. Spindle poles are marked by TBB-2::mCherry (red), the cortex is marked by NMY-2::GFP (green). Dotted white lines indicate boundaries of the dividing AB cells. Scale bar, 10 μm. (**B, C**) Histograms of (**B**) the instantaneous chiral counter-rotating flow velocity $v_c$ (see Materials and methods for definition) and (**C**) the instantaneous contractile flow velocity $v_{contr}$, representing the flow velocity along the long axis

*Figure 4 continued on next page*

*Figure 4 continued*

of the dividing cell (see Materials and methods for definition). Dotted vertical colored lines represent the mean $v_c$ or mean $v_{contr}$, grey boxes represent the error of the mean. Thin black solid lines indicate zero velocities. Inset, blue shading indicates the AB cell analyzed, arrows indicate rotatory cortical flow. (D) Peak rate of spindle skew (filled circles) and peak rate of spindle elongation (open squares) with the RNAi feeding control (control, n = 10; ect-2 (RNAi), n = 12; rga-3 (RNAi), n = 9). The full time series is given in **Figure 4—figure supplement 1**. Errors indicate the error of the mean at 95% confidence.

The online version of this article includes the following figure supplement(s) for figure 4:

**Figure supplement 1.** Spindle elongation and spindle skew dynamics during cytokinesis in the AB cell upon Rho perturbations.

**Figure supplement 2.** Chiral counter-rotating flow and spindle skew quantification for spindle perturbations.

(**Figure 4—figure supplement 2**) indicating that RNAi treatment was effective. Moreover, the AB cell exhibited chiral counter-rotating flows during division ($v_c$ = −5.05 ± 1.12 µm/min) and, although the AB spindle was often misaligned, it skewed substantially during anaphase (spindle skew angle = 23.14 ± 7.46˚, **Figure 4—figure supplement 2**). These results argue against an important role for the known spindle orientation pathway in driving spindle skews. Alternatively, given that the LIN-5-dependent spindle orientation pathway is known to reorient the metaphase spindle in the $P_1$ cell, this pathway might inhibit actomyosin dependent spindle skews in the P/EMS lineage during anaphase. Analyzing spindle skews in $P_1$ upon *lin-5 (RNAi)* and *gpr-1/2 (RNAi)* revealed no significant difference when compared to control embryos, which suggests that the spindle orientation pathway does not overrule actomyosin-dependent spindle skews (**Figure 4—figure supplement 2**). Altogether, we conclude that the LIN-5/GPR-1/2-dependent spindle orientation pathway determines the initial spindle orientation, while chiral actomyosin flows drive the spindle skews observed during anaphase.

If chiral counter-rotating flows indeed drive spindle skews, we predict that a complete absence of chiral flows leads to a complete absence of spindle skews. We test this prediction in the AB cell, by inactivating cortical flows entirely through a fast acting temperature sensitive *nmy-2(ts)* mutant (**Liu et al., 2010**). Worms were dissected to obtain one-cell embryos and mounted at the permissive temperature at 15˚C that allows them to develop normally. Approximately 60 s after the completion of the first ($P_0$) cell division, the temperature was rapidly shifted to the restrictive temperature of 25˚C using a CherryTemp system. Imaging was started as soon as the AB cell entered anaphase. In all *nmy-2(ts)* mutant embryos (12 out of 12), an ingressing cytokinetic ring was absent and cytokinesis failed after temperature shifting, indicating that NMY-2 was effectively inactivated (**Davies et al., 2014**; **Videos 20** and **21**). Interestingly, while the dynamics of spindle elongation, including the peak rate of spindle elongation and the final spindle length, are essentially unchanged as compared to control embryos at restrictive temperature, the average peak rate of spindle skew was significantly reduced (0.33 ± 0.09˚/min as compared to 1.43 ± 0.13˚/min in control embryos, Wilcoxon rank sum test at 95% confidence, **Figure 5A,B**, **Figure 5—figure supplement 1**, **Videos 20** and **21**). This demonstrates that a functional actomyosin cortex with chiral cortical flows is required for spindle skews of the AB cell. In light of our experiments above, we conclude that spindle skews in the AB cell are independent of spindle elongation, but are instead a consequence of the cell skews that are driven by chiral counter-rotating flows. Finally, we performed temperature shift experiments for the first 11 divisions in the nematode, and found that all cells belonging to the AB cell lineage show a significantly reduced skew in the *nmy-2(ts)* embryos at restrictive temperatures as compared to the control (**Figure 5C**). We conclude that a functional actomyosin cortex with chiral cortical flows is required for spindle skews of early AB lineage cell divisions during worm development.

## Left-right asymmetric friction forces can account for the skew of the AB cell

We propose that chiral counter-rotating flows, together with inhomogeneous friction forces with the surroundings, drive cell division skews in a manner similar to a bulldozer rotating on the spot by rotating its chains in opposite directions. Following this analogy, the direction of the cell division skew is determined by (1) the handedness of the chiral counter-rotating flow and (2) the distribution of friction forces across the cellular surface. As described above, the AB cell simultaneously skews in two orthogonal planes, AP-DV and LR-DV. Assuming that there are high friction forces at the AB-$P_1$

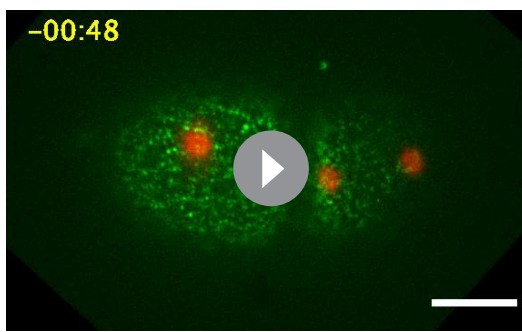

**Video 17.** Cortical flows during the second cell division in *ect-2 (RNAi)* embryos. 4 hr of *ect-2 (RNAi)* leads to reduction in chiral counter-rotating flow velocity (NMY-2::GFP, cortex marked in green and TBB-2::mCherry, spindle poles marked in red) Scale bar 10 μm.
https://elifesciences.org/articles/54930#video17

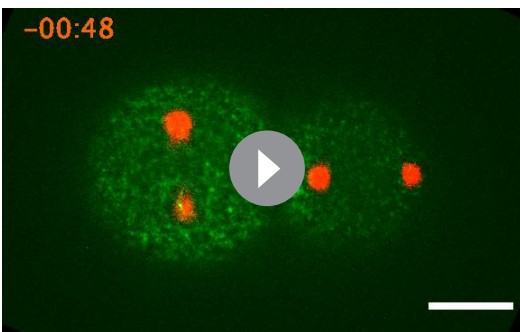

**Video 18.** Cortical flows during the second cell division in *L4440* embryos (feeding control) (NMY-2::GFP, cortex marked in green and TBB-2::mCherry, spindle poles marked in red) Scale bar 10 μm.
https://elifesciences.org/articles/54930#video18

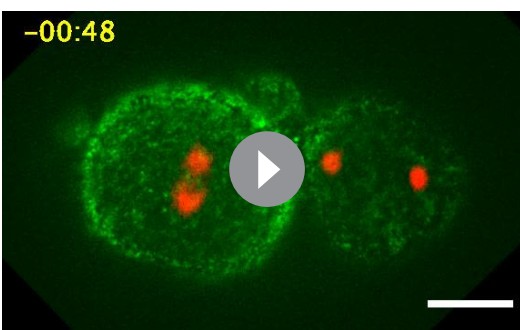

**Video 19.** Cortical flows during the second cell division in *rga-3 (RNAi)* embryos. 6 hr of *rga-3 (RNAi)* leads to increase in chiral counter-rotating flow velocity (NMY-2::GFP, cortex marked in green and TBB-2::mCherry, spindle poles marked in red). Scale bar 10 μm.
https://elifesciences.org/articles/54930#video19

interface, the clockwise skew (when viewed from the anterior down the anteroposterior axis) in the LR-DV plane is consistent with the handedness of the chiral counter-rotating cortical flows (*Figure 3—figure supplement 1*). Similarly, for the observed clockwise skew in the AP-DV plane (when viewed from the left down the left-right axis) to be consistent with the handedness of chiral counter-rotating flows of AB, we expect that the total friction forces between the AB cell and vitelline layer and/or the egg shell are elevated on the right side of the embryo, as compared to the left side (*Figure 5—figure supplement 3*).

In order to test this hypothesis, we first set out to identify signatures of left-right asymmetric friction forces. Because such forces are expected to reduce cortical flow speed, we argued that cortical flow speeds may be different on the future left and right sides of the dividing AB cell. Note that in our analysis, we have so far only reported values of the chiral counter-rotating flow velocity $v_c$ for flows measured on the right side of the embryo. Measuring the chiral counter-rotating flow velocities of the AB cell also on the left side, we indeed find that y-direction flows in each cell half, and consequently $v_c$, are significantly increased on the left side, as compared to the right side (*Figure 5—figure supplement 3*, velocity values are given in *Appendix 1—table 3*). This observation is consistent with the hypothesis that the friction forces experienced by the AB cell are higher on the right side than on the left side.

Based on these qualitative insights, we investigated if the observed spindle skew rates can be quantitatively predicted from measured chiral counter-rotating flow velocities and basic geometric features of the dividing AB cell. This was done in two steps: First, we tested if the cortical flow velocities are consistent with the geometric predictions that follow from the bulldozer analogy. In the second step, we investigated if physical interactions with the egg shell are sufficient to explain the skewing rate and direction.

For the first step, we note that the velocities of the bulldozer chains on top and bottom (corresponding to the left and right side of the embryo) can be measured (A) while standing next to the spinning bulldozer (measurement in the 'lab frame') or (B) while sitting in the co-rotating drivers cab, from which one always expects to observe two perfectly counter-rotating chains. The differences in the velocities measured in A) and B) are due to the overall rotation of the system, that is, the spinning of the bulldozer or, in our case, the skewing of the dividing AB cell in the AP-DV plane. Formalizing this geometric

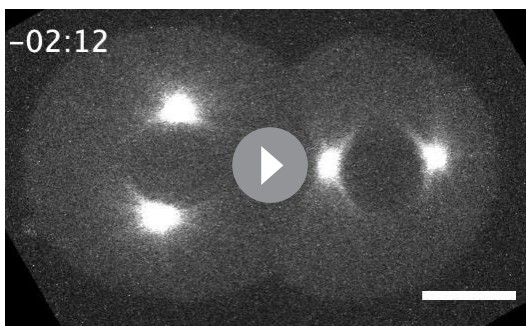

**Video 20.** Spindle skew in the AB cell imaged in control embryos (SWG63) by visualizing TBB-2::mCherry at restrictive temperature of 25°C. Scale bar 10 μm.
https://elifesciences.org/articles/54930#video20

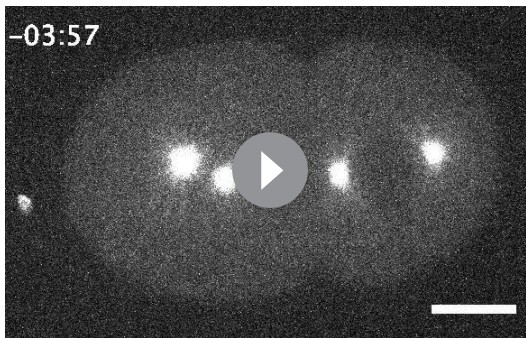

**Video 21.** Spindle skew in the AB cell imaged in the mutant *nmy-2(ts)* embryos (SWG204) by visualizing TBB-2::mCherry at restrictive temperature of 25°C. Scale bar 10 μm.
https://elifesciences.org/articles/54930#video21

description, we derived a parameter-free relation between the skew rate of the dividing AB cell and the cortical y-direction flows on the left and right side (see Appendix). We measured y-direction flows (measurement in the 'lab frame') in wild type, *ect-2 (RNAi)* and *rga-3 (RNAi)* on the left and right side of the embryo and found for all three conditions that the calculated angular skew velocities agree well with the experimentally measured average AP-DV spindle skew rates of the dividing AB cell (*Appendix 1—table 3*). This provides a first, direct quantitative relationship between chiral actomyosin flows and cell division skews. Additionally, it shows that the skewing motion and chiral flows across the AB cell cortex are compatible with a simplified geometry that is analogous to a spinning bulldozer.

In the second step, we used the geometric model to derive the torque balance that connects the AB cell skews to the mechanical interactions with the surrounding. For the skew in the AP-DV plane, we consider a scenario where the relevant torques arise from friction forces between the counter-rotating cell halves and the egg-shell on the left and right side, and then tested, if this is sufficient to explain the observed amplitude and handedness of the spindle skews. Neglecting contributions from the $AB$-$P_1$ cell contact and extra-embryonic fluid that may counteract the AP-DV skew, the torque balance of the system implies an angular skew velocity given by (see Appendix)

$$\omega = \frac{\gamma_L A_L - \gamma_R A_R}{\gamma_L A_L + \gamma_R A_R} \frac{v}{R}. \tag{1}$$

In this expression, $\gamma_L$ and $\gamma_R$ denote local friction coefficients, and $A_L$ and $A_R$ denote the corresponding contact surface areas between the AB cell and the egg shell on the left and right side. The velocity $v$, which is positive for all conditions, is related to the measured chiral counter-rotating flow velocity $v_c$, and $R$ is the distance between the cytokinetic ring and the ROIs where y-direction flows are measured (see Appendix). $\omega < 0$ corresponds to a clockwise skew when the embryo is viewed from the left side. *Equation 1* implies that, if chiral counter-rotating flows are present ($v \neq 0$), any left-right asymmetry in total friction forces $\gamma_L A_L v$ and $\gamma_R A_R v$ leads to an angular skew in the AP-DV plane, while their relative amplitude defines the skew handedness. In order to test this minimal physical model, we estimated the contact surface areas on the left and right side by measuring the area of the cortex that is in focus in our confocal movies ($A_R/A_L \approx 1.3 \pm 0.2$, *Figure 5—figure supplement 3*, *Appendix 1—table 4*). The local friction coefficients were estimated by fitting the thin film active chiral fluid theory to measured flow profiles on the left and right side, where we assumed that the cortical viscosity does not significantly vary across the cell surface and hence that differences in the hydrodynamic length arise from differences in friction coefficients ($\gamma_R/\gamma_L \approx 1.7$, *Figure 5—figure supplement 3*, see Appendix). Together with $R = 4.5\,\mu m$ and $v$ determined from the measured y-direction flows (see Appendix), *Equation 1* allows for each condition to estimate the angular skew velocity that is expected from the torque balance of the system. The resulting angular skew velocities $\omega = -0.42 \pm 0.15°/min$; $-0.28 \pm 0.04°/min$; $-0.49 \pm 0.19°/min$ (*wt; ect-2 (RNAi); rga-3 (RNAi)*) correctly predict the experimentally observed skew handedness and faithfully recapitulate the behavior of the average spindle skew rates in all three conditions (see Appendix). Together, these results lend

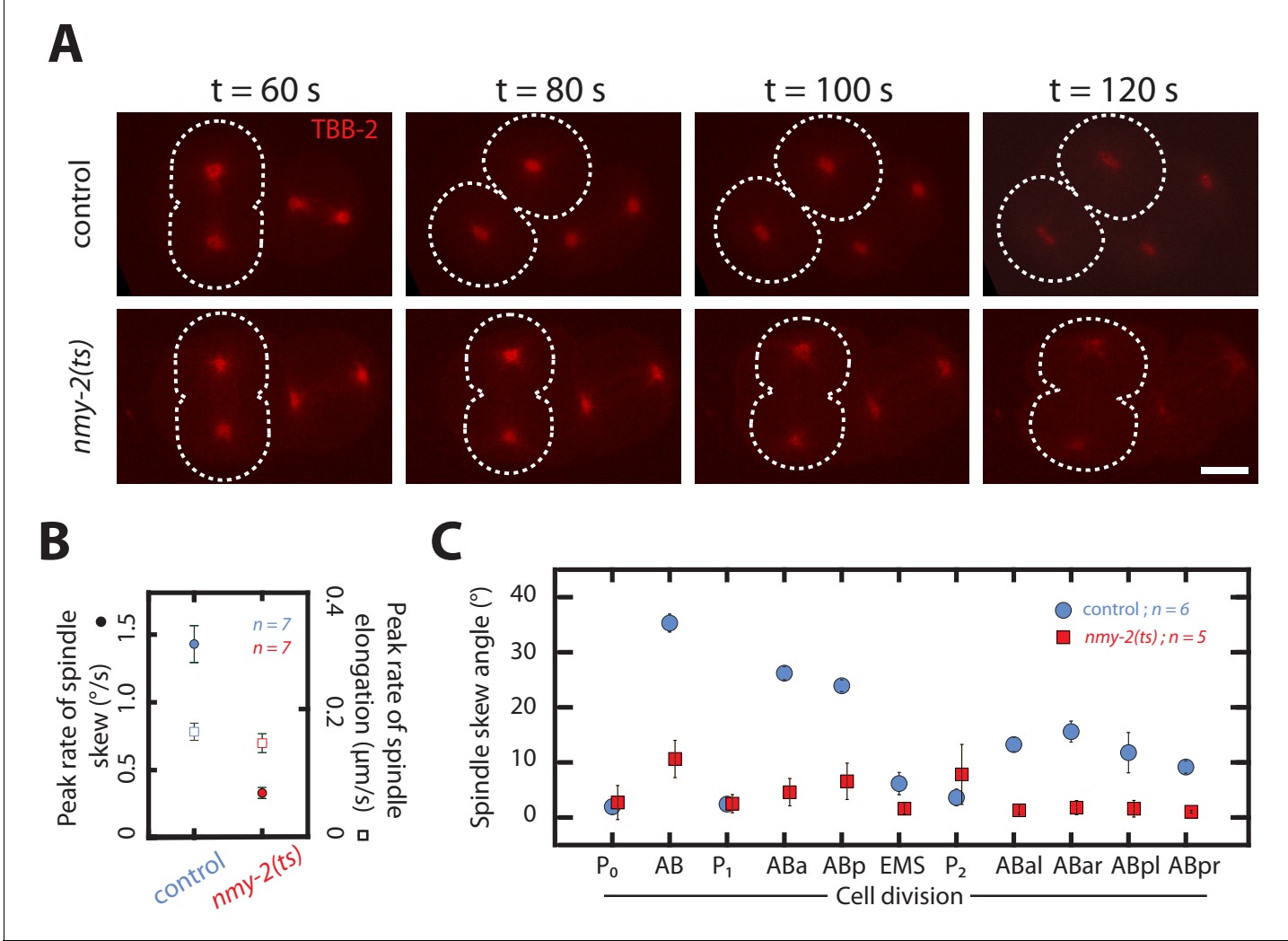

**Figure 5.** Spindle skews in early development are driven by chiral actomyosin flows. (**A**) Fluorescence images of control and myosin temperature-sensitive (ts) mutants (*nmy-2(ts)* ,permissive at 15°C, restrictive at 25°C) two-cell embryos with TBB-2::mCherry labelled for visualizing spindle poles (red). All embryos were shifted from 15°C to 25°C at the end of the first division. Dotted white lines indicate boundaries of the dividing AB cells. Time t = 60 s indicates the onset of cytokinesis. Scale bar, 10 μm. (**B**) Corresponding peak rate of spindle skew (circles) and spindle elongation (squares) for the AB cell in control and *nmy-2(ts)* embryos. The full time series is given in *Figure 5—figure supplement 1*. (**C**) Spindle skew angle in control embryos (blue circles) and *nmy-2(ts)* embryos (red squares) during the first 11 cell divisions at temperature of 25°C. Errors indicate the error of the mean at 95% confidence.

The online version of this article includes the following figure supplement(s) for figure 5:

**Figure supplement 1.** Spindle elongation and spindle skew dynamics during cytokinesis in the AB cell in temperature-sensitive embryos.
**Figure supplement 2.** Chiral counter-rotating flow velocities on the future left side of the dividing AB cell.
**Figure supplement 3.** Parameterization of the minimal model of AB cell skews in the AP-DV plane.

credence to our conclusion that chiral actomyosin flows drive the three-dimensional cell division skew of the AB cell. Furthermore, they suggest that inhomogeneous mechanical interactions with the surrounding are sufficient to explain the handedness and amplitude of these skews.

## Discussion

In this work, we have shown that lineage-specific chiral flows of the actomyosin cortex drive lineage-specific cell rearrangements in early *C. elegans* development. Recently, it was demonstrated that actomyosin flows are required for determining the initial orientation of the mitotic spindle

(*Sugioka and Bowerman, 2018*). Here, we have shown that the actomyosin cortex drives an active counter-rotation of the two halves of dividing early AB lineage cells, leading to cellular repositioning of these cells during anaphase. We propose that chiral counter-rotating flows represent a general mechanism by which dividing cells can reorient themselves in order to allow their daughters to achieve their appropriate positions.

During early *C. elegans* development cell lineage specifications are mainly driven by short-range intercellular signaling cascades (*Priess, 2005*; *Eisenmann, 2005*) and therefore, correct positioning of blastomeres is instrumental for normal development. For example, the cell division skew of the AB blastomere puts the ABp daughter cell in close contact with the posterior $P_2$ cell. In turn, this results in Notch signaling activation in the ABp cell but not in the ABa cell, triggering the distinct development of the ABa and ABp lineages (*Hutter and Schnabel, 1994*). The subsequent spindle skew of ABa and ABp lies at the heart of left-right patterning and is essential for proper left-right asymmetric inductions, again driven by short-range signaling cascades (*Pohl and Bao, 2010*; *Hardin and King, 2008*; *Walston and Hardin, 2006*; *Priess, 2005*). We propose that the chiral flow-driven spindle skews described here, allow cells in the AB lineage to acquire their precise position and thereby permit normal cell fate inductions.

How does the cell lineage determine the presence or absence of chiral cortical flows? We know that AB lineage cells divide symmetrically, while the early P/EMS lineage cell divisions are asymmetric (*Rose and Gönczy, 2014*). We have shown here that the occurrence of chiral counter-rotating actomyosin flows depends on the amount of cortical myosin in each cell half and on the position of the cytokinetic ring. Accordingly, AB lineage cell divisions are symmetric in terms of cytokinetic ring positioning and myosin distribution and display chiral counter-rotating flows, while divisions in the P/ EMS lineage are asymmetric in these measures and chiral counter-rotating flows are absent (*Figure 1B–E*, *Figure 1—figure supplement 4*). Cell lineage determinants like PAR proteins, instrumental for cell polarity, therefore likely act to control chiral flows by modulating the distribution of molecular motors along the cell division axis (*Gross et al., 2019*; *Munro et al., 2004*) .

By which mechanism are the active torques generated that drive chiral counter-rotating cortical flows? We showed previously that chiral active fluid theory can be used to describe the chiral counter-rotating flows during zygote polarization. In this coarse-grained hydrodynamic description, the cortex is treated as a two-dimensional gel that has liquid-like properties and generates active torques. In turn, these torques tend to locally rotate the cortex clockwise (when viewed from the outside) and thereby determine flow handedness. Assuming that the local torque density is proportional to the measured NMY-2 levels, we have shown here that this theory can quantitatively recapitulate the observed chiral flow behavior in the dividing AB blastomeres (*Figure 1—figure supplement 8*). However, the molecular underpinnings that determine the observed flow handedness remain elusive. Additionally, given that many actin-binding proteins localize in a pattern similar to myosin (*Tse et al., 2012*; *Maddox et al., 2005*; *Motegi and Sugimoto, 2006*; *Davies et al., 2018*), and many of these affect the chirality of actomyosin flows in the polarizing one-cell embryo (*Naganathan et al., 2018*), we cannot exclude that other force generators provide the active torque needed for chiral counter-rotating flows. Due to the helical nature of actin, both myosins (*Lebreton et al., 2018*) and formins (*Mizuno et al., 2011*; *Jégou et al., 2013*) can twist actin filaments while they respectively pull or polymerize them, and therefore both are potential sources of molecular torques. Interestingly, myosins as well as formins are also required for numerous cellular and organismal chiral processes (*Tee et al., 2015*; *Davison et al., 2016*; *Kuroda et al., 2016*; *Hozumi et al., 2006*; *Maderspacher, 2016*; *Spéder et al., 2006*; *Chougule et al., 2020*). Future work will be required to identify the molecular source of torque generation in the actomyosin layer, as well as its organization principles that ensure a consistent handedness of the emerging torques and chiral cortical flows.

Our work shows that the handedness of chiral counter-rotating actomyosin flows, together with asymmetries in the effective friction forces between cells and the surrounding determine the direction of cell division skews (*Figure 3A* right panel). In general, such friction forces can vary arbitrarily across the cell surface. Therefore, chiral flow-driven cell division skews represent a three-dimensional process in which skews occur simultaneously in different planes (*Figure 3G*, *Figure 3—figure supplement 1*). Obvious sources of an effective friction force between cell-cell contacts are provided by adhesions. Moreover, our results suggest that differences in friction forces also exist between the AB blastomere and the egg shell on the left and right side. This friction force difference, together

with the handedness of the chiral counter-rotating flow will cause the AB cell to skew clockwise in the AP-DV plane (when viewed from the left side down the left-right axis), thereby establishing the dorsoventral axis. The origin of this difference in friction force may be twofold: A first possibility is that there could be an, as of yet unidentified, adhesion receptor that binds to the vitelline layer and/ or to the egg shell in an inhomogeneous fashion. Although a role for integrins has not been reported during early *C. elegans* development, integrin complexes have been shown to anchor the Tribolium blastoderm to the vitelline envelope (*Münster et al., 2019*). Future work will be required to determine whether integrins could play a similar role in *C. elegans*. A second possibility we favor is that the AB cell surface area that is in contact with the egg shell is different on the left and ride side of the embryo, thus contributing to an asymmetry in the corresponding friction forces. Interestingly, an earlier report showed that the cytokinetic ring of the AB cell ingresses in a left-right asymmetric fashion and pinches off toward the embryo's right side (*Schonegg et al., 2014*). Left/right asymmetric ring ingression likely reduces the AB cell surface area that contacts the egg shell, and – given that ring ingression begins on the left side – this could lead to a left-right asymmetry in contact surface area and thus also an asymmetry in the resulting frictional forces. From our confocal imaging of the actomyosin cortex on left and right sides, we indeed measured a reduced surface area on the left side of the AB cell (*Figure 5—figure supplement 3*, *Appendix 1—table 4*, see Appendix). While these measurements, as well as the fits to approximate the local friction coefficients, only provide coarse approximations, a physical model that incorporates the measured values yields agreement with experimentally measured cell division skews. Hence, our results show that large-scale cellular rearrangements can be accounted for at a quantitative level from the interplay between chiral surface flows and spatial patterns of effective friction forces.

To conclude, our work shows that chiral rotatory movements of the actomyosin cortex are more prevalent in development than one might have suspected. We report on an intricate pattern of chiral flows during early *C. elegans* development, which, together with patterned frictional forces leads to dedicated cell skew and cell reorientation patterns. This is interesting from a physical perspective, since together the cells of the early embryo represent a new class of active chiral material that requires an explicit treatment of angular momentum conservation (*Fürthauer et al., 2013*; *Naganathan et al., 2014*). This is interesting from a biological perspective, since cell skews driven by actomyosin torque generation represent a novel class of lineage-specific morphogenetic rearrangements. Together, our work provides new insights into chiral active matter and the mechanisms by which chiral processes contribute to animal development.

## Materials and methods

### Worm strains and handling

*C. elegans* worms were cultured on NGM agar plates seeded with OP50 as previously described (*Brenner, 1974*). The following *C. elegans* strains were used in this study: LP133 : *nmy-2(cp8[nmy-2:: GFP + unc-119(+)]) I; unc-119(ed3) III* for imaging cortical flows (*Dickinson et al., 2013*). SWG063 : *nmy-2(cp8[nmy-2::GFP + LoxP])I; unc-119(ed3)III; weIs21[Ppie-1::mCherry::beta-tubulin::pie-1 3'UTR]* (cross between (LP133 and JA1559 (Gift from J. Ahringer lab)) for measuring spindle skews in the AB cell. SWG204 : *nmy-2(ne3409ts) I; unc-119(ed3)III; weIs21[pJA138(Ppie-1::mCherry::beta-tubulin:: pie-1 3'UTR)]* (cross between WM179 and JA1559) for temperature-sensitive experiments. TH155 : *unc-119(ed3)III; weIs21[pJA138(Ppie-1::mCherry::beta-tubulin::pie-1 3'UTR)]; PH::GFP* (cross between JA1559 and OD70 [*Kachur and Pilgrim, 2008*]) for measuring spindle skews with the membrane marker and to measure relative cytokinetic ring position along the cell division axis.

### RNA interference

All RNAi experiments in this study were performed at 20°C (*Sijen et al., 2001*). For performing weak perturbation RNAi experiments, L4 staged worms were first incubated overnight on OP50 plates. Young adults were then transferred to respective RNAi feeding plates (NGM agar containing 1 mM isopropyl-$\beta$-D-thiogalactoside and 50 µg ml-1 ampicillin seeded with RNAi bacteria). Feeding times for RNAi experiments were 4–5 hr for *ect-2 (RNAi)* and 6–8 hr for *rga-3 (RNAi)*. For *par-2, chin-1, lgl-1 (RNAi), par-6 (RNAi), par-2 (RNAi), lin-5 (RNAi)* and *gpr-1/2 (RNAi)* conditions, young L4 were transferred to feeding plates 24 hr before imaging. Embryos used as controls for all RNAi

experiments were grown on plates seeded with bacteria containing a L4440 empty vector, and exposed for the same number of hrs as the corresponding experimental RNAi perturbed worms.

The indicated hours of RNAi treatment reflects the time that worm spent on the feeding plate. All dissections were done in M9 buffer to obtain early embryos. Embryos were mounted on 2% agarose pads for image acquisition for the all cell divisions except 4–6 cell stage (*Bargmann and Avery, 1995*). Four cell embryos and uncompressed embryos were mounted using low-melt agarose (*Naganathan et al., 2014*). The *rga-3* feeding clone was obtained from the Ahringer lab (Gurdon institute, Cambridge, United Kingdom) and the *ect-2* feeding clone from the Hyman lab (MPI-CBG, Dresden, Germany). RNAi feeding clones for *par-2, par-6, chin-1 and lgl-1* were obtained from Source Bioscience (Nottingham, United Kingdom).

## Image acquisition

All the imagining done in this study was performed at room temperature (22–23°C) with a spinning disk confocal microscope. Two different spinning disk microscope systems were used for the study. System 1: Axio Observer Z1 - ZEISS spinning disk confocal microscope; Apochromat 63X/1.2 NA objective lens; Yokogawa CSU-X1 scan head; Andor iXon EMCCD camera (512 by 512 pixels) , Andor iQ software. System 2: Axio Observer Z1 - ZEISS spinning disk confocal microscope; Apochromat 63X/1.2 NA objective lens; Yokogawa CSU-X1 scan head ; Hamamatsu ORCA-flash 4.0 camera (2048 by 2048 pixels); micromanager software (*Edelstein et al., 2014*).

Confocal videos of cortical NMY-2::GFP for the first three cell divisions for control and RNAi conditions were acquired using system (1) A stack was acquired using 488 nm laser and an exposure of 150 ms was acquired at an interval of 3 s between image acquisition consisting of three z-planes (0.5 μm apart). The maximum intensity projection of the stack at each time point was then used for further analysis. 21 s of time frames were analyzed for each embryo starting from the time frame when the cytokinetic ring has ingressed ~10%. Time intervals between consecutive frames were increased at later cell stage to avoid photo-toxicity and bleaching. From the 4–6 cell stage onwards, imaging was performed using system (2) A time period of 20 s was analyzed for measuring chiral counter-rotation flow velocity. Image stacks with a spacing of 0.5 μm for 20 z-slices from the bottom surface toward the inside of the embryo were taken with 488 nm laser with exposure of 150 ms at an interval of 5 s during the time of cytokinetic ring ingression and projected with maximum intensity projection.

Imaging of the spindle poles (TBB-2::mCherry) was done using SWG63 (*Figures 2A–B*, *3D, G, H*, *4A* and *5A–C*). TH155 strain was used to image TBB-2::mCherry together with PH::GFP to determine cytokinetic ring position and spindle skew angle in *Figure 2B* for 11-cell stages. SWG204 strain was used for temperature-sensitive experiments (*Figure 4A–C*). Spindle pole imaging for AB cell (TBB-2:: mCherry) was performed by acquiring nine-z-planes (1 μm apart) with a 594 nm laser and exposure of 150 ms on the second system at an interval of 3 s. For long-term imaging using TH155 strain (*Figure 2B*), TBB-2::mCherry together with PH::GFP was imaged in 24 z-slices (1 μm apart) with 594 nm laser and exposure of 150 ms at an interval of 30 s (*Video 9*). For uncompressed embryos, 21 and 25–31 μm z-stacks with z-spacing of 0.5 and 1 μm apart were recorded at time interval between 5 and 10 s using system two for measuring cortical flows and spindle poles during division, respectively. For a faster temporal resolution, we restricted to imaging either spindle skews or cortical flows for individual embryos for all perturbations.

## Image analysis

Cortical flow velocity fields in the 2D $xy$-plane were obtained using a MATLAB code based on the freely available Particle Image Velocimetry (PIVlab) MATLAB algorithm (*Raffel, 2007*). Throughout the study, we used the three-step multi-pass with linear window deformation, a step size of 8 pixels and a final interrogation area of 16 pixels.

To determine the average flow fields that were fitted by the chiral thin film theory (*Figure 1—figure supplement 8*), we first tiled the flow field determined by PIV into nine sections along the long-axis of cells. Within each section, we averaged the flow field over the entire duration during which flows occur. Since we were only interested in flows that are generated by the cytokinetic ring, we excluded embryos in which we could not differentiate whole body rotation from cytokinesis (*Schonegg et al., 2014*). Note that cortical flow velocities change over the duration of cytokinesis

and are likely to be influenced by changes in the AB cell geometry due to asymmetric ring ingression (*Schonegg et al., 2014*). Hence, for quantitative comparisons, we restricted the analysis of flows on the future right side (except for Par polarity perturbations where axis establishment was perturbed) and to the narrow time frame as described in the main text. Flow velocities observed from the future left side have been reported in *Figure 5—figure supplement 3* and in *Appendix 1—table 3*. From all the imaged cell divisions, a small subset of measurements, in which the optical focus on the cortical plane was lost during imaging, was discarded.

From the measured cortical flows, we extracted properties that are later used to quantify their chiral counter-rotating nature in the following way. First, we identified the cleavage plane or the cytokinetic ring by eye. We then defined a region of interest (ROI) on each side of the cytokinetic ring (*Figure 1A*). In order to ensure that PIV measurements were not affected by the saturating fluorescence signal from the cytokinetic ring, we placed the inner boundary of each ROI at a distance of 1 μm from the ring. The position of the outer boundary of each ROI (along the long axis) was scaled with the size of the analyzed cells (10 μm, 8.5 μm, 6 μm, 6.5 μm, 6.5 μm, 6.5 μm for $P_0$, AB, $P_1$, ABa, ABp and EMS, respectively, measured from the position of the cytokinetic ring). For $P_2$, ABal and ABpr cells the distance of the outer ROI boundary to the cytokinetic ring varied around 4 μm, depending on the cell surface area visible in the imaging focal plane. For every cell, the width of ROIs was set equal to the length of the cytokinetic ring visible in the focal plane. In each ROI, we averaged the observed flow field over a period of 21 s. Consistent with earlier findings, we observed a whole cell rotation prior to $P_0$ cytokinesis (*Schonegg et al., 2014*; *Singh et al., 2019*). We discarded a small subset of embryos (4 out of 17), where this whole cell rotation coincided with the time frames in which we performed the flow analysis.

In order to quantify chiral flows, we defined a chiral counter-rotating flow velocity $v_c$ together with a handedness as follows. First, we introduce a unit vector that is orthogonal to the contractile ring and points toward the cell pole that is in the direction of the anteroposterior-axis of the overall embryo ($\underline{e}_x$, see *Figure 1—figure supplement 2*). A second unit vector $\underline{e}_y$ is used, which is parallel to the cytokinetic ring and orthogonal to $\underline{e}_x$. We denote the average velocities determined in the two halves of the cell (measured within the ROIs described above) as $\underline{v}_1$ and $\underline{v}_2$. With this, we define the chiral counter-rotating flow velocity $v_c$ by

$$v_c = \underline{e}_z \cdot (\underline{e}_x \times \underline{v}_2 - \underline{e}_x \times \underline{v}_1), \tag{2}$$

where $\underline{e}_z = \underline{e}_x \times \underline{e}_y$ is the third base vector of a right-handed orthonormal coordinate system. With this definition, $|v_c|$ represents the speed of chiral counter-rotating flows, while the sign of $v_c$ determines the flow's handedness: If cortical flows appear clockwise when viewed from the cytokinetic plane toward each pole, one has $v_c > 0$ and the counter-rotating flows are left-handed. If cortical flows appear counter-clockwise, one has $v_c < 0$ and counter-rotating flows are right-handed. In order to quantify mean chiral counter-rotating velocity, we averaged $v_c$ over multiple embryos for every timepoint that was analyzed. From the analysis of the first nine cell divisions, we found that intensities in the P/EMS lineage are lower and hence the PIV flow analysis is less coherent in the P/EMS lineage as compared to the AB lineage, leading to increased uncertainty in PIV measurements.

To characterize net-rotating flows in a similar fashion, we use the net-rotating flow velocity $v_r$ defined by (*Figure 1—figure supplement 4*)

$$v_r = \underline{e}_z \cdot (\underline{e}_x \times \underline{v}_1 + \underline{e}_x \times \underline{v}_2). \tag{3}$$

If average flows in both cell halves are equal and in the same direction, that is, $\underline{v}_1 = \underline{v}_2$, there are no counter-rotating flows ($v_c = 0$) and the corresponding net-rotating flow is quantified by $v_r$.

Finally, in order to determine a flow measure that captures cortical flows along the cell's long axis and into the contractile ring, we define the contractile flow velocity ($v_{\text{contr}}$) depicted in *Figure 3B* by

$$v_{\text{contr}} = \underline{e}_x \cdot (\underline{v}_2 - \underline{v}_1). \tag{4}$$

For contractile cortical flows into the cytokinetic ring, we have $\underline{e}_x \cdot \underline{v}_1 > 0$ and $\underline{e}_x \cdot \underline{v}_2 < 0$, and therefore $v_{\text{contr}} < 0$.

Flow speed during cell division was calculated by adding the magnitude of mean absolute values of flow vectors in the two boxes from either side of the cytokinetic ring in the dividing cells.

## Spindle skew and elongation analysis

In order to measure spindle skew angles, z-stacks (nine planes, each 1 μm apart recorded in 3 s time intervals) of the spindle poles were first projected on the plane perpendicular to the cytokinetic ring. For uncompressed embryos mounted using low-melt agarose method, z-stacks (25–31 z-planes; 1 μm apart with maximum possible temporal resolution between 5 and 10 seconds time intervals) were captured. For $P_0$, AB and $P_1$ the imaging plane was already perpendicular to the plane of cytokinesis while for the remaining cell divisions the embryos were rotated accordingly using the clear volume plugin (*Royer et al., 2015*) in FIJI. Subsequently, the spindle skew angle was defined as the angle between the line that joins the spindle poles at the onset of spindle elongation (anaphase-B) (initial spindle angle) and at completion of cytokinesis (final spindle angle) (*Figure 2A*). Spindle elongation length was defined as the distance between the two spindle poles during the same timeframes. To measure the dynamics of spindle elongation and skew angle in the AB cell (*Figure 4—figure supplement 1*, *Figure 5—figure supplement 1*) the spindle poles were automatically tracked and the spindle skew angle was plotted using custom-written MATLAB code (refer to spindle skew analysis MATLAB code). In order to compare the dynamics of spindle elongation and spindle skew, we first synchronized all the AB cells in each condition at the onset of anaphase-B and, subsequently, averaged over multiple embryos. The difference beween the spindle angle at each timepoint and the spindle angle at anaphase-B onset timepoint was measured and plotted over time. The maximum value of the slope for spindle elongation and spindle skew angle curves for each embryo between time window of 60–120 s (*Figure 4—figure supplement 1*, *Figure 5—figure supplement 1*) was identified as peak rate of spindle elongation and peak rate of spindle skew. In order to reduce noise in calculating slope, a moving average was performed to smooth the spindle skew angle and elongation profiles using moving slope function in MATLAB. Average peak rates were calculated by averaging the maximum values over multiple embryos for each condition.

## Determining myosin ratio, cytokinetic ring position and fitting myosin profiles

In order to obtain cortical myosin distributions (*Figure 1C*, *Source data 1*), we normalized the axis of cell division and calculated the mean myosin intensity in a 20 pixel-wide stripe along this axis for the same timepoints when $v_c$ was calculated. The myosin ratio (*Figure 1E*) was determined by dividing the mean intensity at the anterior (0–20% of the normalized cell division axis) by the mean intensity at the posterior (80–100% of the normalized cell division axis) for $P_0$, AB, $P_1$ and EMS. For ABa and ABp the myosin intensity was calculated similarly but along the L/R axis. We have restricted myosin intensity measurements to the 20% outer segments of the cell division axis in order to prevent the strong fluorescent signal from the cytokinetic ring from affecting this analysis (*Figure 1—figure supplement 6*). Similar to myosin ratio, myosin concentration at the anterior was normalized with mean posterior myosin concentration for multiple embryos and plotted (*Figure 1—figure supplements 9* and *10*). To determine the position of the cytokinetic ring, we performed midplane imaging using a strain expressing a membrane marker (PH::GFP) and a tubulin marker to label spindle poles (TBB-2::mCherry). The cell division axis was defined as the line through the two opposing spindle poles and was normalized between cell boundaries. The relative cytokinetic ring position was defined as the position where the ingressing ring intersects the normalized cell division axis.

For visualization purposes (*Figure 1C*), we fitted the myosin distributions using the fitting function

$$I(x) = \frac{1}{2}(I_P - I_A)\left(\text{erf}\left[\frac{x - x_r}{w}\right] - 1\right) + I_P + \frac{I_R}{\sqrt{2\pi w^2}}\exp\left[-\frac{(x - x_r)^2}{2 * w^2}\right]. \tag{5}$$

The errorfunction erf and the Gaussian represent contributions from the anteroposterior myosin asymmetry and from the contractile ring, respectively. The fitting parameters $I_P$, $I_A$, $I_R$, $w$ and $x_r$, respectively, describe myosin intensities at the posterior pole, the anterior pole and in the cytokinetic ring, as well as the width and position of the ring.

## Temperature-sensitive experiments

We used the cherry temp temperature control stage from Cherry Biotech for all temperature sensitive experiments in the study. L4 worms carrying *tbb-2::mCherry* (SWG63, control) and *nmy-2(ts); tbb-2::mCherry* (SWG204) were grown at 15°C (permissive temperature for *nmy-2(ts)* mutants) overnight. During image acquisition, embryos were mounted using agar pad mounting method and temperature was maintained using the cherry temp temperature control stage from Cherry Biotech. For imaging the dynamics of spindle elongation and skew angle in the AB cell for *nmy-2(ts)* and control embryos, 60 s after the successful completion of the first cell division ($P_0$ cell division), temperature was raised to 25°C. Embryos were imaged approximately 5 min after the temperature shift to minimize photo toxicity. Similarly, for later cell stages, the temperature shift was carried out right after completion of cytokinesis of the mother cell. A small subset of embryos in which spindle poles of daughter cells could not be visualized were discarded.

## Determining embryo thickness

In order to determine embryo thickness, we collected an ensemble of up to nine two-cell embryos from SWG59, SWG70, OD70 strains mounted using the agar pad method or low-melt agarose method. To measure embryo thickness in utero, young adult worms (24 hrs post L4) were paralyzed in 0.1% tretramisole (Sigma-Aldrich T1512) for 3 min on a cover slip and mounted on 2% agarose pads. The AB cell in the two-cell embryos was projected on a view down the anterior end along the anteroposterior axis. Thickness of the embryo was determined by measuring the end to end distance between the opposing membrane signal (denoted by arrows in *Figure 1—figure supplement 1*). For each embryo, aspect ratio was calculated as the ratio of thickness values of the AB cell when projected along the AP-DV plane and the LR-DV plane as shown in *Figure 1—figure supplement 1*. All embryo rotations for post-processing and time-lapse along various axis were captured using clearvolume as described in the spindle skew analysis section.

# Acknowledgements

We thank Masatoshi Nishikawa for critical inputs on the theory, A Honigmann and P Gönczy for critical reading of the manuscript. We also thank T Hyman and B Bowerman for sharing *C. elegans* strains. We thank T Hyman, Kwabena Badu-Nkansah and all the members of Grill lab for critical input and discussions. We thank Julia Hubatsch for providing movies of *C. elegans* gonads. Some strains were provided by the CGC, which is funded by NIH Office of Research Infrastructure Programs (P40 OD010440). Furthermore, we thank the light microscopy facility of CMCB of TU Dresden, and the light microscopy facility of MPI-CBG for their support. We would like to acknowledge a long standing collaboration with Frank Jülicher on active chiral matter, and thank him for many discussions that were essential for the work here. LP acknowledges funding from the European Union's Horizon 2020 research and innovation program under the Marie Sklodowska-Curie grant agreement No 641639. TCM was supported by the European Molecular Biology Organization (EMBO) long-term fellowship ALTF 1033–2015, and by the Dutch Research Council (NWO) Rubicon fellowship 825.15.010. AM thanks the ELBE Phd fellowship from the Center for Systems Biology, Dresden. SWG was supported by the DFG (SPP 1782, GSC 97, GR 3271/2, GR 3271/3, GR 3271/4), the European Research Council (grants 281903 and 742712).

# Additional information

### Funding

| Funder | Grant reference number | Author |
| --- | --- | --- |
| European Union's Horizon 2020 research and innovation program - Marie Sklodowska-Curie Grant | 641639 | Lokesh G Pimpale |
| EMBO long-term fellowship | ALTF 1033-2015 | Teije C Middelkoop |
| Dutch Research Council (NWO) Rubicon fellowship | 825.15.010 | Teije C Middelkoop |

| DFG | SPP 1782 | Stephan W Grill |
|---|---|---|
| DFG | GSC 97 | Stephan W Grill |
| DFG | GR 3271/2 | Stephan W Grill |
| DFG | GR 624 3271/3 | Stephan W Grill |
| DFG | GR 3271/4 | Stephan W Grill |
| European Research Council | 281903 | Stephan W Grill |
| European Research Council | 742712 | Stephan W Grill |

The funders had no role in study design, data collection and interpretation, or the decision to submit the work for publication.

### Author contributions

Lokesh G Pimpale, performed all experiments, conceptualized the work, analyzed the data and wrote the manuscript; Teije C Middelkoop, assisted with experimental design, conceptualized the work, analyzed the data and wrote the manuscript; Alexander Mietke, developed the theory, conceptualized the work, analyzed the data and wrote the manuscript; Stephan W Grill, S.W.G conceptualized the work, analyzed the data and wrote the manuscript

### Author ORCIDs

Lokesh G Pimpale (iD) https://orcid.org/0000-0001-6145-1516
Teije C Middelkoop (iD) https://orcid.org/0000-0002-0346-6143
Alexander Mietke (iD) https://orcid.org/0000-0003-1170-2406
Stephan W Grill (iD) https://orcid.org/0000-0002-2290-5826

### Decision letter and Author response

Decision letter https://doi.org/10.7554/eLife.54930.sa1
Author response https://doi.org/10.7554/eLife.54930.sa2

## Additional files

### Supplementary files

- Source data 1. Data values that have been depicted in the figures of the manuscript.
- Source code 1. MATLAB code for tracking spindle skews.
- Transparent reporting form

### Data availability

We have provided all data in the manuscript and supporting files. Due to the large size of the movies, the source movies are available upon request.

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

## Appendix 1

## Thin film active chiral fluid theory

Cortical flows in *Caenorhabditis elegans* embryos have been successfully described using the thin film theory of active chiral fluids (**Naganathan et al., 2014**; **Naganathan et al., 2016**). We use the fact that cells are approximately axisymmetric, and we assume that the active forces during cell division vary mostly along their symmetry axis, in the following referred to as *long axis*. In this case, the hydrodynamic equations of a chiral thin film can be formulated as an effectively one-dimensional system of equations, capturing flows parallel ($v_x$) and orthogonal ($v_y$) to the long axis:

$$\eta \partial_x^2 v_x + \partial_x T = \gamma v_x \tag{S1}$$

$$\frac{\eta}{2} \partial_x^2 v_y + \partial_x \tau = \gamma v_y. \tag{S2}$$

Here, $\eta$ is the viscosity of the cortex and $\gamma$ is the friction with surrounding material. Note that the assumption of a homogeneous friction $\gamma$ is a strong simplification, where we neglect details of the potentially inhomogeneous mechanical interactions between cells, as well as with the extra-embryonic fluid and the egg shell. Gradients of active tension $T$ and active torques $\tau$ lead to contractile and chiral flows, respectively. It has been demonstrated previously that active tension and active torques both dependent on the local myosin concentration in the cortex (**Mayer et al., 2010**; **Naganathan et al., 2014**; **Naganathan et al., 2016**). Using the fluorescent myosin intensity $I(x)$ as a proxy for the myosin concentration, we can write in the simplest case $T = T_0 I(x)$ and $T = \tau_0 I(x)$.

Denoting the length of the measurement domain by $L$ (approximately the length of the cell's long axis), we can identify three independent parameters in the model **Equations S1 and S2**: The characteristic contractile and chiral velocities $v_T = LT_0/\eta$ and $v_\tau = L\tau_0/\eta$, respectively, as well as the hydrodynamic length $\ell = \sqrt{\eta/\gamma}$.

## Fitting cortical flows

For the fitting of cortical flows for given myosin profiles, we closely follow our previous work (**Naganathan et al., 2014**). Briefly, we use experimentally determined myosin profiles $I(x)$ in **Equations S1 and S2** to calculate flows $v_x$ and $v_y$. We then determine the parameters $v_T$, $v_\tau$ and $\ell$ for which the predicted flows best match the experimental data. To extract unique solutions from **Equations S1 and S2**, we use the experimentally measured flow velocities at the boundaries of the measurement domain as boundary conditions. Note that due to imaging limitations for smaller cells (after the eight-cell stage) as development progresses, it was not possible to extract spatially resolved velocity line profiles and fits for the cortical flows in ABal cell and ABpr cell. The final best fit parameters for AB, ABa and ABp cells are shown in **Appendix 1—table 1**. The corresponding contractile flow profiles $v_x$ and chiral flow profiles $v_y$ are shown in **Figure 1—figure supplement 8**.

**Appendix 1—table 1.** Best fit parameters. The average domain lengths (approximately the size of the cell's long axis) on which the myosin profiles were given were $L_{AB} = 27.4\,\mu m$, $L_{ABa} = 19.5\,\mu m$ and $L_{ABp} = 17.9\,\mu m$. The corresponding fits are shown in **Figure 1—figure supplement 8**.

| Cell type | $\ell$ [μm] | $v_T$ [μm/min] | $v_\tau$ [μm/min] |
|---|---|---|---|
| AB | 18.7 | 0.15 | 0.25 |
| ABa | 10.4 | 0.19 | 0.15 |
| ABp | 8.2 | 0.4 | 0.4 |

We also considered asymmetrically dividing cells ($P_0$, $P_1$, EMS) in which, instead of counter-rotating flows ($|v_c|>0$, $v_r \approx 0$), mainly net-rotating flows occur ($v_c \approx 0$, $|v_r|>0$) (*Figure 1—figure supplement 4*). Following the same fitting procedure as described in the previous section, we noticed that the chiral thin film theory *Equations S1 and S2* generally could not account quantitatively for the observed cortical flow profiles. However, using the theory it is still possible to rationalize qualitative properties of the observed cortical flows, as we discuss in the following.

## Qualitative properties of myosin distributions and chiral flows

While the chiral thin film theory does not recapitulate all of the experimentally observed flows quantitatively, the theory allows linking qualitative predictions to key properties of observed myosin distributions and chiral flows. In particular, we noticed that the myosin profiles in asymmetrically diving cells consistently featured a rather asymmetrically positioned contractile ring (*Figure 1D*). Furthermore, the myosin profiles of asymmetrically dividing cells are asymmetric with respect to the cytokinetic ring and exhibit plateaus toward the anterior cell poles (*Figure 1C* and *Figure 1E*). An overview of these qualitative properties for the P/EMS lineage and the AB-lineage is given in *Appendix 1—table 2*.

**Appendix 1—table 2.** Qualitative properties of myosin distributions and chiral flows consistently observed in P/EMS lineage ($P_0$, $P_1$, EMS, $P_2$) and AB-lineage (AB, ABa, ABp, ABal, ABar) cells.

|  | Myosin peak position | Anterior-posterior myosin | Chiral flow properties |
|---|---|---|---|
| AB-lineage | Centered | Symmetric | $v_r \approx 0$, $|v_c|>0$ |
| P/EMS lineage | Off-center | Asymmetric | $|v_r|>0$, $v_c \approx 0$ |

To establish how a contractile ring and an overall myosin asymmetry generally affect flows predicted by the chiral thin film theory, we generalize the chiral thin film equations (*Naganathan et al., 2014*) to curved surfaces and solve them on an ellipsoid using corresponding synthetic torque profiles (*Figure 1—figure supplement 7*). We find that a symmetrically placed ring pattern yields perfectly counter-rotating flows ($v_r = 0$, $|v_c|>0$, *Figure 1—figure supplement 7*, left), while anterior-posterior myosin asymmetry ratio results in net-rotating flow ($|v_r|>0$, $v_c = 0$, *Figure 1—figure supplement 7*, middle) while an anterior-posterior myosin asymmetry along with a symmetrically placed ring pattern yields net-rotating flows such as observed in P/EMS cells (*Figure 1—figure supplement 7*, right). To study the combined effect of a varying position of the contractile ring and a myosin asymmetry, we develop in the following section a simple minimalistic model of the system.

## Chiral flow minimal model

In the following, we develop a minimal description of the occurrence of counter- and net-rotating flows based on key characteristics of the myosin profiles discussed in the previous section. We consider a simplified myosin profile in the form

$$I(x) = I_r(x) + I_{ap}(x) \tag{S3}$$

with

$$I_r(x) = I_R w \delta(x - x_r) \tag{S4}$$

$$I_{ap}(x) = (I_P - I_A)\Theta(x - x_r) + I_A. \tag{S5}$$

Here, $\delta(x)$ is the delta function, $\Theta(x)$ the Heaviside step function and $x_r$ is the position of

an idealized cytokinetic ring. The intensity profile $I_r(x)$ captures the presence of a myosin peak (the later contractile ring) of width $w$ and average intensity $I_R$ at $x_r$. The profile $I_{ap}(x)$ represents the large-scale asymmetry in myosin between anterior pole ($I_A$) and posterior pole ($I_P$). We solve **Equation S2** on a domain of length $L$ and for boundary conditions $v_y(0) = v_y(L) = 0$ using the simplified myosin intensities given in **Equations S4 and S5**, which yields

$$\frac{v_y^{(i)}}{v_\tau} = \begin{cases} A_{(i)} \sinh(\alpha x/L) & \text{for } 0 \leq x \leq x_r \\ B_{(i)} \sinh[\alpha(x-L)/L] & \text{for } x_r \leq x \leq L \end{cases}. \tag{S6}$$

Here, $\alpha = \sqrt{2\gamma L^2/\eta}$ is an inverse dimensionless hydrodynamic length and $v_\tau = L\tau_0/\eta$ the characteristic velocity associated with active torques. The coefficients in **Equation S6** are given by

$$A_r = I_R \frac{w}{L} \frac{\cosh[\alpha(1-x_r/L)]}{\sinh\alpha} \qquad\qquad A_{ap} = \frac{I_P - I_A}{\alpha} \frac{\sinh[\alpha(1-x_r/L)]}{\sinh\alpha}$$

$$B_r = I_R \frac{w}{L} \frac{\cosh(\alpha x_r/L)}{\sinh\alpha} \qquad\qquad B_{ap} = -\frac{I_P - I_A}{\alpha} \frac{\sinh(\alpha x_r/L)}{\sinh\alpha}.$$

With these coefficients, **Equation S6** describes flows $v_y^r$ resulting from the contractile ring given in **Equation S4** and flows $v_y^{ap}$ resulting from the asymmetric myosin profile given in **Equation S5**. Note that, as expected, flows due to the presence of the contractile ring vanish for $I_R = 0$ and flows due to the presence of a myosin asymmetry vanish if $I_A = I_P$. General flow profiles are given as superposition of the two contributions: $v_y = v_y^r + v_y^{ap}$.

Finally, to evaluate these solutions, we define a dimensionless counter-rotating velocity $\tilde{v}_c$ and the net-rotating velocity $\tilde{v}_r$ in analogy to the quantities introduced in the main text and methods, **Equations 2 and 3**, as

$$\tilde{v}_c = \frac{1}{L} \left( \int_{x_r}^{x_r+\Delta} \tilde{v}_y \, dx - \int_{x_r-\Delta}^{x_r} \tilde{v}_y \, dx \right) \tag{S7}$$

$$\tilde{v}_r = \frac{1}{L} \int_{x_r-\Delta}^{x_r+\Delta} \tilde{v}_y \, dx. \tag{S8}$$

Here, $\tilde{v}_y = v_y/v_\tau$ and we consider a window of 15 % ($\Delta/L = 0.15$) of the total length toward either side of the idealized contractile ring over which the mean flow velocity is determined (**Figure 1A**, **Figure 1—figure supplements 2 and 4**). Using this minimal model, we can now investigate how the combined contributions of a large-scale myosin asymmetry ($I_{ap}$) and a varying contractile ring position ($I_r$) affect the counter-rotating velocity $\tilde{v}_c$ and net-rotating velocity $\tilde{v}_r$. In particular, we fix for the discussion $\alpha = 1$ and $w/L = 0.1$, as well as $I_R/I_P = 2$. In this case, the counter-rotating flow velocity $\tilde{v}_c$ and the net-rotating flow velocity $\tilde{v}_r$ are only functions of the relative ring position $x_r/L$ and the anterior-posterior myosin ratio $I_A/I_P$ with properties shown in **Figure 1—figure supplement 7** and described in the following.

For an anterior-posterior symmetric myosin profile, counter-rotating flows $|\tilde{v}_c|$ have a weak maximum in their amplitude if the contractile ring is at a centered position $x_r/L = 0.5$ (**Figure 1—figure supplement 7**). Furthermore, a centered contractile ring has no effect on the counter-rotating flow measure $\tilde{v}_c$ if a large-scale myosin asymmetry $I_A/I_P > 1$ is introduced (**Figure 1—figure supplement 7**). However, for an off-centered ring $x_r/L > 0.5$ the presence of a myosin asymmetry will contribute flows that reduce $\tilde{v}_c$ and can even lead to vanishing counter-rotating flows $\tilde{v}_c = 0$ if $I_A/I_P$ is sufficiently large. While this matches the qualitative experimental observations listed in **Appendix 1—table 2**, the required asymmetry predicted by the theory is significantly larger than the experimentally measured one.

Also, the behavior of $\tilde{v}_r$ in our minimal model is compatible with the qualitative properties listed in **Appendix 1—table 2**. In particular, for a large-scale myosin asymmetry $I_A/I_P > 1$, the net-rotation velocity $\tilde{v}_r$ is negative for any ring position, while it can becomes positive for $x_r/L > 0.5$ if $I_A = I_P$ (**Figure 1—figure supplement 7**). Furthermore, net-rotating flows vanish

for a symmetric myosin profile $I_A = I_P$ and a centered contractile ring $x_r/L = 0.5$ (**Figure 1—figure supplement 7**), which corresponds to the observations made in AB-lineage cells. Finally, the presence of large-scale myosin asymmetries $I_A/I_P \neq 1$ is generally expected to contribute to net rotating flows (**Figure 1—figure supplement 7**). This holds true for essentially arbitrary positions of the contractile ring and indicates that such asymmetries could play an important role for developing a better understanding of net-rotating flows in the P/EMS lineage cells in the future.

## Theoretical descriptions of AB cells skews

In the following, we introduce a minimal model that can quantitatively link cell skews to the chiral cortical flows and mechanical interactions with the surrounding. Note that while the model describes the general scenario of two confined, counter-rotating surfaces, we focus its application to the case of the dividing AB cell and its skew in the AP-DV plane, for which all the required experimental information is available to make quantitative predictions.

We consider an idealized configuration of two connected incompressible surfaces that represent the cortices of the future ABa and ABp cell during AB cell division and reorientation-skew. Chiral counter-rotating cortical flows correspond to rotations of these surfaces in opposite directions around a common axis. For simplicity, we neglect more complex aspects of the surface deformations that occur during the cell division and skewing inside the constraining egg shell. This system is analogous to the two chains under a bulldozer, where the motion of the chains represent chiral cortical actomyosin flows and the operators cab, with a fixed orientation relative to the two chains, indicates the orientation of the dividing AB cell. The two connected surfaces are additionally confined by two opposing rigid surfaces that are parallel to the AP-DV-midplane and represent the egg shell (**Figure 5—figure supplement 3**).

## Geometric description of counter-rotating and skewing surfaces

We first describe the surface motion in a reference frame that is co-rotating with the angular skew occurring in the AP-DV-plane. We assume that active chirality leads to a counter-rotating surface motion, and the local velocities of the ABa and ABp cell on the the embryo's future *left* side at a distance $R$ from the cytokinetic ring are given by $-v_a$ and $v_p$, respectively. Observing the two counter-rotating surfaces in a common co-rotating reference frame, the local velocities of each cell on the future *right* side are then given by $v_a$ and $-v_p$, respectively (**Figure 5—figure supplement 3**). In experiments, cortical flows $(v_L^{(a)}, v_L^{(p)}, v_R^{(a)}, v_R^{(p)})$ of the ABa and ABp cell on each surface are measured in the lab frame, which is defined as the reference frame in which the egg shell is at rest. These measurements are related to the parameters of the simplified geometric model by

$$v_L^{(a)} = -v_a + \omega_g(R - \delta) \tag{S9}$$

$$v_L^{(p)} = v_p - \omega_g(R + \delta) \tag{S10}$$

$$v_R^{(a)} = v_a + \omega_g(R - \delta) \tag{S11}$$

$$v_R^{(p)} = -v_p - \omega_g(R + \delta). \tag{S12}$$

Here, $\omega_g$ is the angular skew velocity around an axis that is orthogonal to the AP-DV plane and $\delta$ describes a possible displacement of the rotation axis from the center (**Figure 5—figure supplement 3**). If $\delta > 0$ ($\delta < 0$) the axis of rotation is shifted toward the ABa (ABp) cell. The sign in front of $\omega_g$ is chosen such that $\omega_g > 0$ ($\omega_g < 0$) corresponds to anti-clockwise

(clockwise) skews when viewed from the left side down the left-right axis (*Figure 5—figure supplement 3*). From *Equations S9–S12*, we then find

$$\omega_g = \frac{1}{2R}(\alpha - \beta) \tag{S13}$$

$$\delta = R\frac{\alpha + \beta}{\beta - \alpha} \tag{S14}$$

$$v_a = \frac{1}{2}\left(v_R^{(a)} - v_L^{(a)}\right) \tag{S15}$$

$$v_p = \frac{1}{2}\left(v_L^{(p)} - v_R^{(p)}\right), \tag{S16}$$

where $\alpha = (v_L^{(a)} + v_R^{(a)})/2$ and $\beta = (v_L^{(p)} + v_R^{(p)})/2$. Note that *Equations S9–S16* essentially represent the transformation of velocities between two reference frames that rotate relative to each other.

To test if this simplified geometric representation of two counter-rotating and skewing surfaces is consistent with experiments, we measured $v_L^{(a)}$, $v_L^{(p)}$, $v_R^{(a)}$ and $v_R^{(p)}$ during the late state of the AB cell skew and computed the cell skew parameters using *Equations S13–S16* for the *wt*, *ect*-2 and *rga*-3 conditions (*Appendix 1—table 3*). For all three conditions, the calculated angular skew velocities $\omega_g$ agree well with the average angular skewing velocities of the spindle measured in experiments (*Appendix 1—table 4*, right column). This strongly suggests that chiral cortical flows across the cortex are not only qualitatively, but even quantitatively linked to the corresponding skewing motion of the spindle. Furthermore, we find for all three conditions $\delta/R \approx 1$ (*Appendix 1—table 3*), which implies that the axis around which the AB cell skews is shifted toward the ABa cell. During later stages of the skew, the time point at which the velocities $v_L^{(a)}$, $v_L^{(p)}$, $v_R^{(a)}$ and $v_R^{(p)}$ given in *Appendix 1—table 3* were measured, this can indeed be observed as a feature of the spindle skew (*Figure 5—figure supplement 3*): Both spindle poles of the AB cell move nearly parallel to the AP axis, but the ABp cell's spindle pole does so faster. As a result, the rotation axis is located closer to the ABa spindle pole than to the ABp spindle pole.

**Appendix 1—table 3.** Chiral counter-rotations in experiments and in a minimal geometric model. Left: Experimentally measured chiral cortical flows ($\pm$ error at 95 % confidence interval) of the ABa and ABp cell at late stages of the spindle skew for different conditions. Flows on both cells are measured on the future left ($v_L^{(a)}$, $v_L^{(p)}$) and right side ($v_R^{(a)}$, $v_R^{(p)}$). Right: Assuming two rigid, counter-rotating and skewing surfaces, the velocities measured in the lab frame (left) can be mapped via *Equations S13–S16* to the skewing properties ($\omega_g$, $\delta$) and surface velocities ($v_a$, $v_p$) in the co-rotating frame ($\pm$ propagated uncertainty). Distance between cytokinetic ring and points of velocity measurements: $R = 4.5\,\mu m$.

| Measured vel. | *wt* | *ect*-2 | *rga*-3 | Equations S13–S16 | *wt* | *ect*-2 | *rga*-3 |
|---|---|---|---|---|---|---|---|
| $v_L^{(a)}$ [μm/min] | $-4.6 \pm 0.8$ | $-3.0 \pm 0.7$ | $-5.9 \pm 0.7$ | $\omega_g$ [°/s] | $-0.31 \pm 0.14$ | $-0.19 \pm 0.18$ | $-0.38 \pm 0.15$ |
| $v_R^{(a)}$ [μm/min] | $5.0 \pm 0.7$ | $2.6 \pm 1.0$ | $5.3 \pm 0.9$ | $\delta/R$ | $1.14 \pm 0.13$ | $0.80 \pm 0.42$ | $0.83 \pm 0.13$ |
| $v_L^{(p)}$ [μm/min] | $8.6 \pm 0.7$ | $5.7 \pm 0.7$ | $9.8 \pm 0.6$ | $v_a$ [μm/min] | $4.8 \pm 0.8$ | $2.8 \pm 0.85$ | $5.6 \pm 0.8$ |
| $v_R^{(p)}$ [μm/min] | $-2.4 \pm 0.5$ | $-2.5 \pm 1.0$ | $-3.2 \pm 0.6$ | $v_p$ [μm/min] | $5.5 \pm 0.6$ | $4.1 \pm 0.85$ | $6.5 \pm 0.6$ |

**Appendix 1—table 4.** Comparison of the angular velocity $\omega$ determined from the physical skew model with experimentally measured average spindle skew rates. The angular velocity $\omega$ is calculated for each condition using *Equations S21* with $R = 4.5\,\mu m$ and $v = (v_a + v_p)/2$ (see

*Appendix 1—table 3*). $\Gamma$ is given by *Equation S23*, with the hydrodynamic lengths $\ell_L = 24.8\,\mu m$ and $\ell_R = 18.7\,\mu m$ (*Figure 5—figure supplement 3*), and the measured contact surface areas on the future left and right side given by $A_L = 166.13 \pm 22.63\,\mu m^2$ and $A_R = 210.49 \pm 39.9\,\mu m^2$, respectively. Theory error: $\pm$ propagated uncertainty, Measurement error: $\pm$ 95 % confidence interval

| Condition | Angular skew velocity $\omega$ [°/s] from theory | Measured average spindle skew rate [°/s] |
|---|---|---|
| wt | $-0.42 \pm 0.15$ | $-0.32 \pm 0.04$ |
| ect-2 | $-0.28 \pm 0.04$ | $-0.29 \pm 0.04$ |
| rga-3 | $-0.49 \pm 0.19$ | $-0.36 \pm 0.16$ |

## Derivation of a physical skew model

Chiral surface flows and the motion of the AB cell relative to the surrounding material can in general give rise to friction forces and external torques that act on the cell. For the angular cell skew in the AP-DV-plane, the relevant frictions forces would have to occur dominantly through interactions with the egg shell on the future left and right side of the embryo (*Figure 5—figure supplement 3*). The corresponding flows are described by the velocities $v_L^{(a)}$, $v_L^{(p)}$, $v_R^{(a)}$ and $v_R^{(p)}$ measured in the lab frame, such that the external torques on future left and right side are given, respectively, by

$$
\begin{aligned}
\tau_L &= R\left(F_L^{(a)} - F_L^{(p)}\right) \\
&\approx -R\gamma_L\left(A_L^{(a)} v_L^{(a)} - A_L^{(p)} v_L^{(p)}\right) \\
&= R\gamma_L\left[A_L(v - \omega R) + \omega\delta\left(A_L^{(a)} - A_L^{(p)}\right)\right]
\end{aligned}
\tag{S17}
$$

$$
\begin{aligned}
\tau_R &= R\left(F_R^{(a)} - F_R^{(p)}\right) \\
&\approx -R\gamma_R\left(A_R^{(a)} v_R^{(a)} - A_R^{(p)} v_R^{(p)}\right) \\
&= -R\gamma_R\left[A_R(v + \omega R) + \omega\delta\left(A_R^{(p)} - A_R^{(a)}\right)\right].
\end{aligned}
\tag{S18}
$$

Here, $F_L^{(a)} = -\gamma_L A_L^{(a)} v_L^{(a)}$ depicts the total external friction force on the future left side of the ABa cell (similarly for the other forces), $A_L = A_L^{(a)} + A_L^{(p)}$ and $A_R = A_R^{(a)} + A_R^{(p)}$ are the total contact surface areas between the AB cell and egg shell on the left and right side (*Figure 5—figure supplement 3*), respectively, and we have used *Equations S9–S12* for simplicity with $v_a = v_p = v$. The external torques in *Equation S17 and S18* transfer a total torque

$$
\tau_c = \tau_L + \tau_R
\tag{S19}
$$

onto the AB cell. This total torque on the other hand is balanced by a dissipative torque $\tau_\omega$ that arises through the angular skew of the dividing AB cell inside the egg shell, that is,

$$
\tau_\omega + \tau_c = 0.
\tag{S20}
$$

For simplicity, we consider $\tau_\omega = -\bar{\eta} R^3 \omega$, where $\bar{\eta}$ denotes an effective rotational viscosity that captures interactions of the skewing AB cell with the extra-embryonic fluid inside the egg shell and potentially also interactions with the $P_1$ cell. Combining *Equations S17–S20*, we find

$$
\omega = \Gamma \frac{v}{R},
\tag{S21}
$$

with

$$
\Gamma = \frac{A_L \gamma_L - A_R \gamma_R}{\bar{\eta} R + \gamma_L\left(A_L + \frac{\delta}{R}\Delta A_L\right) + \gamma_R\left(A_R + \frac{\delta}{R}\Delta A_R\right)},
\tag{S22}
$$

where $\Delta A_L = A_L^{(p)} - A_L^{(a)}$, $\Delta A_R = A_R^{(p)} - A_R^{(a)}$. *Equation S21* with $\Gamma$ given in *Equation S22* provides a minimal model that can be used to estimate the expected angular cell skew velocity $\omega$ from the measured cortical velocities $(v_L^{(a)}, v_L^{(p)}, v_R^{(a)}, v_R^{(p)})$ that define $v$, basic known measures of the cell geometry $(A_L, A_R, R)$ and the friction parameters $(\gamma_L, \gamma_R, \bar{\eta})$.

## Quantitative comparison of the physical skew model with experiments

In the following, we want to use *Equation S21* together with the experimentally measured parameters listed in *Appendix 1—table 3* to calculate the expected angular skew frequency $\omega$. Note that the agreement of $\omega_g$ calculated from *Equation S13* with the experimental values validates the simplifying geometric assumptions of our model, while *Equation S21)* indicates how specific mechanical interactions with the surrounding can lead to the handedness and amplitude of the angular skew.

To compute the required coefficient $\Gamma$ given in *Equation S22*, we use that $\delta/R \approx 1$ (*Appendix 1—table 3*) for all conditions, and we consider approximately similar contact surface areas of the ABa and ABp cell halves on each side $(A_L^{(a)} \approx A_L^{(p)}, A_R^{(a)} \approx A_R^{(p)})$. In this case, $\Gamma$ only depends on the total contact surface areas $A_L$ and $A_R$ that we estimate from confocal images $(A_R/A_L \approx 1.3$, *Figure 5—figure supplement 3*). Furthermore, we can express the friction coefficients in *Equation S22* as $\gamma_L = \eta_L/\ell_L^2$ and $\gamma_R = \eta_R/\ell_R^2$, where $\ell_L$ and $\ell_R$ are the hydrodynamic lengths. The latter are determined by fitting the chiral thin film theory *Equations S1 and S2* to measured flow profiles on the left and right side of dividing AB cells (*Figure 5—figure supplement 3*), from which we find $(\ell_L/\ell_R)^2 = \eta_L\gamma_R/(\eta_R\gamma_L) \approx 1.7$. Note, that we can in general not exclude spatial variations of the cortical viscosity that would lead to $\eta_L \neq \eta_R$ and accordingly affect the estimated local friction parameters. However, even if viscosity variations would fully account for the observed difference in hydrodynamic length such that $\gamma_L = \gamma_R$, the contact area asymmetry with $A_R > A_L$ would still be sufficient to explain the handedness of the observed skews. Finally, to allow for quantitative predictions, we make the simplifying assumption $\eta_L \approx \eta_R = \eta$, such that *Equation S22* is for all three conditions given by

$$
\begin{aligned}
\Gamma &= \frac{A_L/\ell_L^2 - A_R/\ell_R^2}{\bar{\eta}R/\eta + A_L/\ell_L^2 + A_R/\ell_R^2} \\
&\approx \frac{A_L\ell_R^2 - A_R\ell_L^2}{A_L\ell_R^2 + A_R\ell_L^2}.
\end{aligned}
\tag{S23}
$$

In the second step, we have assumed that the cortical viscosity $\eta$ is large compared to the effective viscosity $\bar{\eta}$ that describes interactions with the extra-embryonic fluid $(\bar{\eta}R/\eta \approx 0)$. *Equation S21* with $\Gamma$ given in *Equation S23* is equivalent to *Equation 1* and shows that contributions to the hydrodynamic length and to the contact surface area that are different on the future left and right side can generate an angular cell skew, where their relative values define the handedness of the skew. With *Equation S23*, the angular skew velocity $\omega$ given in *Equation S21* is fully determined by parameters known from experiments and the corresponding values for all three conditions are shown in *Appendix 1—table 4*. Despite the geometric simplifications, our minimal model can predict the experimentally measured cell skew velocities for all three conditions within a relative error margin of approximately 5–25%.

