## [Decision Letter]

**Acceptance summary:**

This study demonstrates that acto-myosin driven intracellular flows are lineage specific during *C. elegans* development and that they re-orient mitotic spindles during the division of cells. Using theory, the authors are able to explain the quantitative observations made in their experiments. This work supports the notion that developmentally controlled intracellular flows play an important role for determining cell fate.

**Decision letter after peer review:**

Thank you for submitting your work entitled "Cell lineage-dependent chiral actomyosin flows drive cellular rearrangements in early development" for consideration by *eLife*. Your article has been reviewed by two peer reviewers, and the evaluation has been overseen by a Reviewing Editor and a Senior Editor. The reviewers have opted to remain anonymous.

Our decision has been reached after consultation between the reviewers. Based on these discussions and the individual reviews below, it was concluded that in order to satisfy the reviewers' concerns, additional experiments will be required that are expected to take more than 2 months, i.e. the maximum period envisaged by *eLife* for a revision. We therefore regret to inform you that your work will not be considered further for publication in *eLife* at this stage.

While the reviewers found that your study was interesting and well written, they were not convinced that this manuscript represents a sufficient advance over your previous work published in *eLife* in 2014 (Naganathan et al). The specific major points of criticism that were identified in the discussion between the reviewers were:

1) Technical concern:

The reviewers were not convinced that the measured spindle skew was due to torque-driven spindle rotation. If true rotation is observed, it was considered important to clarify that it is not due to compression of the embryo. The reviewers considered control experiments essential. For example, "hanging drop" imaging may resolve this.

2) Concerns over the proposed mechanism:

The reviewers criticized that the mechanism of how actin-myosin counter-flows induce spindle tilt/skew remains unclear. More importantly, they were not convinced that the study demonstrates that counter-flows were necessarily the cause of spindle skew/tilt, as they saw several other possibilities that were not considered in the manuscript:

i) The known active spindle orientation pathway in P1 cells might remove spindle skew. This could be tested by disrupting spindle orientation pathways.

ii) Cell length extension, overall cortical contractility in combination with geometrical confinement instead of cortical flows may cause spindle skew. This could be tested for example by using embryos without eggshell.

iii) The reviewers saw also the possibility that cortical flows change cell shape/positioning (by frictional coupling of flow to the neighboring cells/egg shell) instead of cell-internal frictional coupling of the spindle to the cortex. Analysis of cell shape changes under normal or perturbation conditions could give indications.

Please find below the original comments of the reviewers. We hope that they will be useful for your next steps with this manuscript.

Reviewer #1:

Pimpale et al. report on the quantitative description of rotary actomyosin flows during early cell divisions in C.elegans development and their role in spindle skew and cell positioning.

This study builds on previous work, especially the article by Naganathan et al., 2014, in which the presence of chiral flows and a theoretical model description based on thin film active chiral fluid theory have been introduced. The authors now continue to employ elegant quantitative live cell imaging and theoretical modelling to show that chiral counter-rotating flows are present in the following nine cell divisions of C.elegans development. Lineage specific analysis revealed that chiral counter-rotating flows are restricted to the AB lineage (versus P/EMS lineage) and a minimal theoretical model is presented to qualitatively describe the appearance of chiral counter-rotating versus net rotating-flows observed in different cell lineages. The authors use various complementary perturbation approaches to show that rotary cortical flows correlate with the positioning of the cell division axis. These observations suggest that rotary cortical flows generate active forces to drive cell re-positioning in multicellular tissues, establishing an interesting concept with potential relevance in various developmental and dynamic tissue remodeling processes.

The presented article is very well written, provides high-quality data and analysis methods and includes a relevant discussion of obtained results.

General concerns and questions are related to the following points:

The authors describe that chiral counter-rotating flow velocities in the AB lineage decrease as development progresses. Does this decrease in vc occur because cortical flow is reduced in the e_y_ direction or absolute flow velocities are reduced?

Is decreasing rotary flow also associated with a change in absolute myosin density levels in dividing cells?

Why was the myosin ratio (Figure 1E) determined only from intensity values within the first 20% outer segments along the cell division axis?

The authors show that rotary cortical flows are correlated with spindle skew and cell positioning but the underlying mechanism remains to be addressed. Can the authors dissect if cortical flows induce cellular turning by flow-friction with the environment (neighboring cells/egg shell) followed by spindle skew, or is spindle positioning primarily mediated by cell internal friction between the rotating cortex and the spindle and subsequent reorientation of the cell division axis? Interfering with cell-cell/cell-matrix adhesion or cortex-spindle coupling could allow to distinguish these scenarios.

A minimal theoretical model is presented how rotary cortex flows depend on cleavage furrow position and myosin density asymmetries in dividing cell halves. How far can these parameters be considered as independent variables considering possible feedback between the cell cortex dynamics and furrow positioning? (see also Sedzinski et al., 2011).

The authors provide a discussion on how active torques are generated on a molecular level. It would be interesting to further comment on how the handedness of chiral flow could be generated in first instance.

Supplementary figure 1C shows a pronounced dependence of chiral velocity vc over time in the AB cell. Does this velocity profile correlate with changes in myosin ratios or cleavage furrow positioning according to model predictions?

Supplementary figure 3C provides 2 exemplary modelling results for rotary flows on an ellipsoid surface. As the authors are studying rotary flows during division it would be interesting to provide additional examples with settings such as asymmetric myosin densities in the presence of high myosin levels in the cleavage furrow. Also, an animation video showing how rotary flows change depending on cleavage furrow positioning and varying myosin density levels in dividing cells would be very instructive to illustrate changes in cortical flow dynamics under varying parameters of furrow position and myosin density asymmetries.

Reviewer #2:

The manuscript by Pimpale et al. investigates the mechanisms by which actomyosin chiral flows drive spindle skew and consequently the positioning of cells within the early *C. elegans* embryo.

In Naganathan et al., 2014, the authors demonstrated (1) that the acto-myosin cortex generates active torque, (2) that this torque can result in counter rotation of AB daughter cells (ABa, ABp) during cytokinesis, and (3) that the resulting spindle skew breaks left-right symmetry resulting in a relative shift in the position of blastomeres on the right and left sides of the embryo.

This new work is largely an extension of the previous paper, adding two key findings that are well supported by the experimental data: First, that chiral counter-rotating flows are not restricted to the ABa/p cells analysed previously, and second, that counter-rotating flows are restricted to symmetrically-dividing AB lineage cells and absent in the asymmetrically dividing P/EMS lineage cells, pointing to lineage-specific regulation. The authors then use a combination of theory and experiment to understand the origin of this lineage specific behaviour, suggesting the key determinants are furrow position and myosin asymmetry. However, in my opinion, these latter aspects of the manuscript are not sufficiently explored to support their conclusions, particularly with respect to analysis division of AB and P1 which constitute the majority of Figures 2-4.

Major Concerns:

1) While their 2014 work nicely illustrates the importance of counter-rotation in left-right symmetry-breaking and presents a model for understanding the emergence of chiral flows, this manuscript primarily documents that counter-rotations are not confined to the previously studied ABa/p cells. The relevance of counter-rotations for later divisions is unclear. Notably the degree of counter-rotation and resultant skew declines after ABa/p, suggesting it may simply be an echo of processes required in Aba/p. Is it clear whether the embryo cares about counter rotations beyond ABa/p? Similarly, rotation of AB relative to P around the AP axis, while interesting in defining the physical mechanism, would seem to be irrelevant to the embryo given that the left-right/DV axes are not yet defined.

2) One major issue is that the manuscript uses the division of AB (not the well characterized ABa/p from their prior work) to validate many of their conclusions regarding the link between counter-rotation and spindle skew (e.g. Figures 2-4). However, in all cases, spindle skew in AB is analysed in the X-Y plane (see Figure 2), whereas counter-rotation of the dividing AB cell relative to P ought to primarily induce skew in the Y-Z plane (e.g. rotation around the AP axis), which would only be evident in an end-on view down the A-P axis. How counter-rotation around the cleavage furrow would lead to the observed tilting of the division axis towards the P cell in X-Y is not obvious, unless I am misunderstanding the geometry.

3) Have the authors considered potential issues of embryo compression? Prior works have shown that embryo compression by agarose pads can influence developmental processes. Most critical for this work, the spindle in AB invariably comes to lie in the X-Y plane in agarose compressed embryos, even if initially oriented along z-axis, presumably due to geometric constraints. This may limit the ability of the authors to reliably detect the contribution of counter-rotating flows to spindle skew by fixing the plane of cell division. Presumably, if there was a counter-rotation in AB and the division plane was fixed by eggshell geometry, one might expect to see rotation of P1 relative to AB. Other groups have used so-called hanging drops to avoid this issue and facilitate end on analysis. I wonder if this geometry would be more useful if one wants to properly analyse spindle skew in AB.

4) The authors use par mutants to assess the link between fate, chiral rotations, and spindle skew. As predicted, in par mutants that yield two AB like cells, both cells undergo counter-rotating flows and high levels of spindle skew, while mutants yielding two P1 like cells show no chiral rotations and reduced spindle skew. However, P1 cells have an active spindle orientation pathway which may dominate the system, preventing analysis of actomyosin generated spindle skew. I also worry about this analysis due to concerns in point 2 above – tilt in X-Y seems very different from spindle skew that would be driven by counter-rotation. Finally, it is unclear to me what kind of skew to expect when one obtains two adjacent cells with similar counter-rotations (e.g. Figure 2D), which I would have expected might just cancel out if divisions are all symmetric.

5) The authors nicely show how counter-rotation depends on myosin activity (though this is known from their prior work and already explored extensively for Aba/p cells). Notably, at low levels of myosin, counter-rotation declines, which they then explore in Figure 3 and 4. However, the link to spindle skew is tenuous for the AB cell. Could one not explain the data more simply by imagining that cell length extension and the contractility of the cortex during cytokinesis combined with eggshell geometry drives tilt? A more compliant / soft cortex might simply deform during division and hence tilt less, while a stiff/contractile cortex would require the dividing cell to tilt as cell poles encounter the eggshell. This would also be compatible with the coordinated tilting of the two identical AB like cells produced by par mutants, which seems hard to explain from the balance of chiral flows alone.

6) The proposed model suggests that furrow position and myosin asymmetry are the key differentiating features of P lineage cells that prevent counter-rotating flow. However, the other obvious difference is that the two lineages show dramatic differences in actomyosin contractility and as the authors show myosin activity is directly related to the magnitude of counter-rotation. Can the authors exclude that reduced myosin activity in P lineage cells, rather than ring position and myosin asymmetry, is the key difference? The model makes two key predictions that the authors do not test: First, a symmetrically-positioned cleavage furrow should lead to counter-rotating flow independent of myosin asymmetry – hence P_0_ cells with spindle positioning defects should exhibit counter-rotating flow. Second, polarized, asymmetrically dividing cells such as P_0_ should never show chiral counter-rotation regardless of actomyosin levels. What happens in P_0_ if actomyosin activity is enhanced, e.g. with a strong rga-3/4 (RNAi), which the authors showed in 2014 enhances chiral flows in the one cell embryo?

7) The authors make a strong point that given symmetric myosin, there is nevertheless a weak influence of furrow position. However, this plot (Sup Figure 3E) is somewhat misleading as the maximal effect on rotation is ~1% if myosin is symmetric – one could as easily argue that there is an insignificant contribution of furrow position relative to other factors.

[Editors’ note: further revisions were suggested prior to acceptance, as described below.]

Thank you for submitting your article "Cell lineage-dependent chiral actomyosin flows drive cellular rearrangements in early development" for consideration by *eLife*. Your revised manuscript has been seen by the two original reviewers of your manuscript, an additional reviewer who is a senior expert in the field with relevant expertise, a Reviewing Editor and Aleksandra Walczak as the Senior Editor. While all reviewers see potential for an excellent paper, the majority opinion still sees major deficits as outlined below.

The Reviewing Editor has drafted this decision to help you prepare a revised submission.

As the editors have judged that your manuscript is of interest, but as described below that additional experiments/analysis/modelling/theory are required before it is published, we would like to draw your attention to changes in our revision policy that we have made in response to COVID-19 (https://elifesciences.org/articles/57162). First, because many researchers have temporarily lost access to the labs, we will give authors as much time as they need to submit revised manuscripts. We are also offering, if you choose, to post the manuscript to bioRxiv (if it is not already there) along with this decision letter and a formal designation that the manuscript is "in revision at *eLife*". Please let us know if you would like to pursue this option. (If your work is more suitable for medRxiv, you will need to post the preprint yourself, as the mechanisms for us to do so are still in development.)

Summary:

All reviewers appreciate that the analysis of the P_0_ cell has been improved and new data using spindle positioning mutants have been added that are consistent with the model. The reviewers agree also that the data generally support the idea that chiral flows can induce spindle rotation (which was known from previous work) and that this study shows that this operates also in other cells, making it a more general phenomenon. It is also appreciated that the manuscript presents a theory that can explain chiral flows.

It remained however unclear to the reviewers how chiral flow around the long axis of the cell produces the tilt in X-Y. Given that so much of the analysis rests on this tilt phenomenon, as indeed does the claim to relevance for development, because the manuscript suggests that this tilt allows ABp-P2 contact, the absence of an attempt to provide a physical explanation for the tilt was considered a critical weakness. The reviewers expect at least some attempt to understand theoretically how the observed tilt can be obtained from the patterns of flow that is observed.

Essential revisions:

The reviewers note that the motion of an elongated objects in a rotational flow is one of the best-studied problems in viscous fluid mechanics and conclude that even with the obvious complexities in the present setup (geometry, non-Newtonian fluids, etc.) it should be possible to make some sensible estimates.

In this context, a particular statement that the reviewers struggled with was "Strikingly, when viewed from either embryo pole, the closest AB-like cell skews clockwise, meaning that the two juxtaposed cells counter- rotate (new, Figure 2—figure supplement 2) and new Video 14–16." The reviewers argued that the direction of rotation of both cells in this particular case can be explained by the cell-cell contact area having an increased friction coefficient when compared to the cell- vitelline membrane and/or cell-eggshell contact. Given this topology, the handedness of the chiral flow indeed would be expected to give rise to clockwise rotation of both cells when viewed from the closest pole." This raises the question of whether the cells counter-rotate or both rotate clockwise. The reviewers also note that the observed X-Y tilt of the cells in the same direction has not been addressed here.

Finally, the notion emerged that the manuscript is not particularly reader-friendly, a concern that should be addressed. It is recommended to add a clear schematic early on that explains the geometry more clearly than Figure 1 (or the supplements). Moreover, the terminology "no chiral counter-rotating flows" was found to be confusing. Does that mean that there are chiral flows but they are not counter-rotating, or there are counter-rotating flows but they are not chiral? Indeed, chiral in this sense is not well-explained geometrically (here a schematic could help), especially since the experiments appear to only look at planar projections of the flows.

We would also like to encourage you to include the species studied in the title.

In conclusion, we regret to inform you that your manuscript cannot be accepted in its present form. Because the reviewers see however potential for a manuscript that can be accepted at *eLife*, they encourage you to address the concerns raised above. In that case, we would be prepared to consider a re-revised version of your manuscript.

---

## [Author Response]

[Editors’ note: The authors appealed the original decision. What follows is the authors’ response to the first round of review.]

We do want to point out that in comparison to the first series of papers published on chiral actomyosin movements, we here report on the discoveries that chiral flows in the

actomyosin cortex are much more prevalent that perhaps envisioned, that they arise in

approximately half of the early divisions of the nematode, and that they do so in a lineage-specific manner. Hence, chiral flows are part of normal development, and they are under specific control. What are these flows utilized for? Through an elegant combination of RNAi experiments as well as utilizing rapid-switching temperature-sensitive mutants, and together with physical theory, we show that chiral flows of actomyosin drive cell rearrangements and cause cell shifts, a morphogenetic activity that in C. elegans had previously been attributed to the mitotic spindle. We think that this changes to a significant degree our understanding of early C. elegans development, and casts actomyosin cellular function in the context of development into a new perspective.

Again, we acknowledge the points that have been raised by the reviewers. We have

amongst the authors carefully discussed the major concerns and have in the meantime also already carried out and analyzed a number of additional experiments that have been requested by the reviewers. For example, by imaging uncompressed embryos (using a method equivalent to the hanging drop method) we have now confirmed that embryo compression does not affect counter rotating flows or overall spindle skew movements, addressing a key technical concern. We also already have the data ready in which we perturbed components of the known spindle orientation pathway (lin-5 and gpr-1/2). The results argue against a role for known spindle orientation pathways in driving chiral skews. Importantly, the known spindle reorientation pathway in the P_1_ cell acts early during mitosis (prophase) while the spindle skew events that we report here act much later, during anaphase (Rose & Kemphues, Development, 1998). To rule out an inhibitory role of the spindle reorientation pathway in P_1_ we will in the coming weeks perform RNAi treatment against let-99 and quantify the cell division skew during anaphase B. The results will be added in the revised manuscript. Together, we are now confident that we can address and answer all but one of the reviewer concerns within the two month period, and within the next six weeks.

With respect to the experiment that we cannot do within the two months, we will not be

able to rule out the possibility that cell length extension, combined with contractility and geometric confinement, drives cell division skew (reviewer 2, concern 5). However, we do show that subtle reduction of the RhoA regulators ect-2 and rga-3 specifically alters the chiral counter rotation velocity and cell division skew (fig 4b), while leaving the non-chiral component of the cortical flow unaffected (fig 4c). This strongly suggest that contractility and cortical stiffness are not significantly affected upon mild depletion of RhoA regulators. This argues that actomyosin contractility per se does not play a major role in driving cell division skews, and lends credence to the statement that active torques and chiral flows do. However, it is not possible to completely decouple the effects, but we hope that you agree with us that this does not matter that much, as we do provide significant evidence that chiral flows play an essential role here.

Reviewer #1:Pimpale et al. report on the quantitative description of rotary actomyosin flows during early cell divisions in C.elegans development and their role in spindle skew and cell positioning.This study builds on previous work, especially the article by Naganathan et al., 2014, in which the presence of chiral flows and a theoretical model description based on thin film active chiral fluid theory have been introduced. The authors now continue to employ elegant quantitative live cell imaging and theoretical modelling to show that chiral counter-rotating flows are present in the following nine cell divisions of C.elegans development. Lineage specific analysis revealed that chiral counter-rotating flows are restricted to the AB lineage (versus P/EMS lineage) and a minimal theoretical model is presented to qualitatively describe the appearance of chiral counter-rotating versus net rotating-flows observed in different cell lineages. The authors use various complementary perturbation approaches to show that rotary cortical flows correlate with the positioning of the cell division axis. These observations suggest that rotary cortical flows generate active forces to drive cell re-positioning in multicellular tissues, establishing an interesting concept with potential relevance in various developmental and dynamic tissue remodeling processes.The presented article is very well written, provides high-quality data and analysis methods and includes a relevant discussion of obtained results.General concerns and questions are related to the following points:The authors describe that chiral counter-rotating flow velocities in the AB lineage decrease as development progresses. Does this decrease in vc occur because cortical flow is reduced in the e_y_ direction or absolute flow velocities are reduced?Is decreasing rotary flow also associated with a change in absolute myosin density levels in dividing cells?

To address question, we have now performed additional quantifications of flow velocities and myosin intensities in AB, ABa and ABp cells. We indeed found that both, 1) the absolute flow velocity and 2) the overall myosin intensities, are smaller in ABa/ABp than in AB cells. This suggests that an overall reduction in myosin density during the developmental progress also contributes to the reduction in the observed chiral flows.

We have added these results in three new supplementary figure panels (Figure 1—figure supplement 5 A-C) and complemented the Results section in the main text accordingly (paragraph three).

Why was the myosin ratio (Figure 1E) determined only from intensity values within the first 20% outer segments along the cell division axis?

This choice was made in order to prevent the strong fluorescent signal from the cytokinetic furrow to affect the measurements of the myosin ratio. During the onset of cytokinesis, the time point at which we measure the myosin ratio, the cytokinetic furrow often covers up to 60% of the cortical surface that is in the focal plane and consequently, we used only myosin intensities from the outer 20% of the AP-axis.

In order to better clarify this point, we have now added a new supplemental figure panel with a new schematic (Figure 1—figure supplement 6) and we have added a clarification in the Materials and methods section (subsection “Determining myosin ratio, cytokinetic ring position and fitting myosin profiles:”).

The authors show that rotary cortical flows are correlated with spindle skew and cell positioning but the underlying mechanism remains to be addressed. Can the authors dissect if cortical flows induce cellular turning by flow-friction with the environment (neighboring cells/egg shell) followed by spindle skew, or is spindle positioning primarily mediated by cell internal friction between the rotating cortex and the spindle and subsequent reorientation of the cell division axis? Interfering with cell-cell/cell-matrix adhesion or cortex-spindle coupling could allow to distinguish these scenarios.

We thank the reviewer raising this important point. We propose that the cell division skew is due to chiral cortical flows in combination with an inhomogeneous friction of the rotating cell with its surrounding. In our work, we have used the orientation of the spindle only as a quantitative measure for the orientation of the dividing cell and our results suggest that neither a force generation within the spindle nor the interaction between the spindle and the cortex, are the main drivers of the skew.

To exclude a role of spindle-cortex interactions in guiding the spindle orientation, we quantified both the cortical flow profile and the cell division skew of AB cells upon RNAi perturbation of *lin-5,* which is known to affect cortex spindle interactions (Srinivasan et al., 2003; Galli et al., 2011). Upon perturbation of *lin-5,* we observed spindle defects that have been previously described, including a misaligned metaphase spindle in the P_1_ cell (new, Figure 3—figure supplement 2C). However, chiral actomyosin flows were still present in in the AB cell (new, Figure 3—figure supplement 2B). Although the orientation of the AB spindle was often aberrant, a pronounced cell axis rotation (new, Figure 3—figure supplement 2A) along with the spindle skews could still be observed (new, Figure 3—figure supplement 2F). These results argue against direct spindle-cortex interactions, but suggest that the spindle simply follows the cell axis rotation and that the main driver of these rotations are the chiral cortical flows.

We have compiled the aforementioned *lin-5 (RNAi)* results in the new, Figure 3—figure supplement 2 and complemented the main text accordingly (subsection “Cells exhibiting chiral counter-rotating actomyosin flows also undergo spindle skews” and “ Chiral counter-rotating flows drive spindle skews”).

A minimal theoretical model is presented how rotary cortex flows depend on cleavage furrow position and myosin density asymmetries in dividing cell halves. How far can these parameters be considered as independent variables considering possible feedback between the cell cortex dynamics and furrow positioning? (see also Sedzinski et al., 2011)

We thank the reviewer for pointing this out. Indeed, several biological mechanisms exist that could couple the cleavage furrow position to myosin asymmetries. These include, for example, polar relaxation (Sedzinski et al., 2011) PAR polarity that drives polar myosin asymmetry (Gross et al., 2018, Nat. Physics) or signaling between an asymmetrically positioned mitotic spindle and the cortex (Grill et al., 2001).

While these phenomena may constrain the parameter-space of a fully quantitative description, we used our minimal model only to gain qualitative insights into the relative effects of contractile ring position and myosin asymmetry. Furthermore, a possible coupling between ring position and myosin asymmetry would not change the quantitative part of the theoretical analysis. To predict cortical flows, we directly use the observed myosin profiles which already contain the information about ring position and myosin asymmetry – as given inputs. Therefore, the quantitative flow predictions are independent of the more complex processes that set up the global myosin profile.

We have adjusted the text in the Results (paragraph three) to clarify these points.

The authors provide a discussion on how active torques are generated on a molecular level. It would be interesting to further comment on how the handedness of chiral flow could be generated in first instance.

We thank the reviewer for raising this interesting point. While this is in fact a matter of currently ongoing research in our lab, we have now complemented the discussion and included our current conceptual view on how the handedness of the flow could arise on a molecular level. Briefly, we argue that chiral flow handedness originates from 1) the action of molecular torque generator and 2) the orientation of this torque generator to generate in plane torque density.

We have added a part in the Discussion to further elaborate on this matter (paragraph four).

Suppl. Figure 1C shows a pronounced dependence of chiral velocity vc over time in the AB cell. Does this velocity profile correlate with changes in myosin ratios or cleavage furrow positioning according to model predictions?

We thank the reviewer for this interesting question. To test this, we have now quantified myosin profiles over time and did in fact not observe changes in the myosin ratio or the furrow position during early (t=1s), mid (t=24s) or late cytokinesis (t=45s) (new, Figure 1—figure supplement 3). During cytokinesis progression, the myosin gradient sharpens strongly along the contractile ring, and the ring starts ingressing. Ingression makes it move out of the focal plane, which prevents a more quantitative and spatially resolved flow analyses after t = 30 seconds (new Figure 1—figure supplement 2D, right). However, during this additional analysis, we noted that the total flow speed changes over time in a similar fashion as vc, which explains a large part of the temporal changes of counter-rotating flows. Moreover, we expect measurements of vc to be affected by the changes in the overall AB cell geometry and by the furrow ingression, which are both very dynamic in this time window.

In light of this, we have now added a new panel to supplementary figure (new, Figure 1—figure supplement 3) and discuss these aspects in the manuscript in (Results paragraph two)

Supplementary figure 3C provides 2 exemplary modelling results for rotary flows on an ellipsoid surface. As the authors are studying rotary flows during division it would be interesting to provide additional examples with settings such as asymmetric myosin densities in the presence of high myosin levels in the cleavage furrow. Also, an animation video showing how rotary flows change depending on cleavage furrow positioning and varying myosin density levels in dividing cells would be very instructive to illustrate changes in cortical flow dynamics under varying parameters of furrow position and myosin density asymmetries.

We agree with the reviewer that these additional examples would be insightful to add.

We have now added a new sub panel in Figure 1—figure supplement 7A; right, to depict the rotatory flow pattern that emerges when a contractile ring and a myosin asymmetry are simultaneously present. As suggested by the reviewer, we have also attached several animations that illustrate the effects of different myosin profiles on the chiral flow fields:

1) Increasing myosin asymmetry in the presence of a centered contractile ring (Video 6)

2) Varying position of the contractile ring in the presence of symmetric myosin. (Video 7)

3) Varying position of the contractile ring in the presence of asymmetric myosin. (Video 8)

We have also included discussed these results in the Results section (paragraph five). We also further elaborate on this point in Results section with new experiments (paragraph seven).

Reviewer #2:[…] Major Concerns:1) While their 2014 work nicely illustrates the importance of counter-rotation in left-right symmetry-breaking and presents a model for understanding the emergence of chiral flows, this manuscript primarily documents that counter-rotations are not confined to the previously studied ABa/p cells. The relevance of counter-rotations for later divisions is unclear. Notably the degree of counter-rotation and resultant skew declines after ABa/p, suggesting it may simply be an echo of processes required in Aba/p. Is it clear whether the embryo cares about counter rotations beyond ABa/p? Similarly, rotation of AB relative to P around the AP axis, while interesting in defining the physical mechanism, would seem to be irrelevant to the embryo given that the left-right/DV axes are not yet defined.

The reviewer raises the important point that spindle skew events of the AB cell or the daughter cells of ABa/ABp may not be functionally relevant for embryonic development. However, early work from the Schnabel lab (Hutter and Schnabel, 1994) has shown that the skew of the dividing AB cell puts the ABp daughter cell in close contact with the posterior P_2_ cell. In turn, this results in Notch signaling activation in the ABp cell but not in the ABa cell, which is essential for normal development of the ABp and ABa lineages. Given that most intercellular signaling cascades during early development are short range (Hardin and King, 2008; Priess, 2005, Wormbook; Walston and Hardin, 2006), we propose that subtle changes in daughter cell positioning, and thereby intercellular contact areas, can be instrumental for cell lineage determination, also beyond the 6-cell stage.

We have added a discussion on this topic to the Discussion section (paragraph two).

2) One major issue is that the manuscript uses the division of AB (not the well characterized ABa/p from their prior work) to validate many of their conclusions regarding the link between counter-rotation and spindle skew (e.g. Figures 2-4). However, in all cases, spindle skew in AB is analysed in the X-Y plane (see Figure 2), whereas counter-rotation of the dividing AB cell relative to P ought to primarily induce skew in the Y-Z plane (e.g. rotation around the AP axis), which would only be evident in an end-on view down the A-P axis. How counter-rotation around the cleavage furrow would lead to the observed tilting of the division axis towards the P cell in X-Y is not obvious, unless I am misunderstanding the geometry.

We thank the reviewer for raising this point. We agree with the reviewer that one would expect the AB cell to skew in the YZ-plane and that tilting towards the P cell in the XY-plane seems not obvious. When analyzing the cell division skew in an end-on view down the A-P axis, we did not observe obvious skew in the YZ-plane (new, Figure 3—figure supplement 1). However, as the reviewer correctly points out in his next comment (3), we used an embryo mounting method that slightly compresses the embryo (referred to as mildly squeezed embryos; new, Figure 1—figure supplement 1) and this likely prevents cell division skew in the YZ-plane. Therefore, in order to address this comment, as well as comment 3, we mounted embryos by embedding them in low-melt agarose to prevent embryo compression.

We first confirmed that this method indeed resulted in embryos being uncompressed by measuring the aspect ratio. The aspect ratio was 1.02 ± 0.3 under these conditions, while the aspect ratio was 1.29 ± 0.14 when using the classic agar pad method (new, Figure 1—figure supplement 1). Subsequently, we analyzed spindle skews in the AB cell in both the XY- and YZ-plane. We found that the spindle skew in the XY-plane was comparable in compressed and uncompressed embryos and therefore our key results are not affected by embryo compression (new Figure 2B and 2G). Moreover, as was predicted by the reviewer, we did observe an additional clockwise skew in the YZ-plane, when viewed end-on down the AP axis, only in uncompressed embryos (new Figure 2G and new, Figure 2—figure supplement 1). Similarly, in our previous work (Naganathan et al., 2014) we had shown that, the skew of the ABa and ABp cells also occurs in 2 perpendicular planes: the well-known skew is in the AP-LR plane, but we have also reported a substantial skew in the DV-LR plane, perpendicular plane.

Together, these findings show that chiral flow-driven skew of the cell division axis is a complex process that occurs in 3 dimensions. We propose here that, in addition to chiral surface flows, an inhomogeneous friction of the cell surface with its surroundings will determine the direction of the skew. Therefore, predicting the skew in 3 dimensions would involve quantitative mapping of both the cortical flow as well as friction coefficients along the entire surface of the dividing cell. While such an analysis is beyond the scope of this study, it indeed provides an interesting challenge for future research.

We now have added these additional findings in the main text (Resutls section) along with a new sub panel in main figure 3 (Figure 3G) and added two new supplementary figures (new, Figure 1—figure supplement 1 and new, Figure 3—figure supplement 1) along with new, Video 10. In addition, we added a part in the Discussion to elaborate more on the cell division skews in three dimensions (see Discussion paragraph five).

3) Have the authors considered potential issues of embryo compression? Prior works have shown that embryo compression by agarose pads can influence developmental processes. Most critical for this work, the spindle in AB invariably comes to lie in the X-Y plane in agarose compressed embryos, even if initially oriented along z-axis, presumably due to geometric constraints. This may limit the ability of the authors to reliably detect the contribution of counter-rotating flows to spindle skew by fixing the plane of cell division. Presumably, if there was a counter-rotation in AB and the division plane was fixed by eggshell geometry, one might expect to see rotation of P_1_ relative to AB. Other groups have used so-called hanging drops to avoid this issue and facilitate end on analysis. I wonder if this geometry would be more useful if one wants to properly analyse spindle skew in AB.

We thank the reviewer for these suggestions. As discussed in detail in our response to point 2, above, we have performed an analysis on uncompressed embryos and found that the skew in the XY plane is similar in compressed and uncompressed conditions. Therefore, our conclusions are not affected by embryo compression. Furthermore, we have found an additional skew in the YZ plane that was masked by the embryo compression (new, Figure 2—figure supplement 1).

We have added these new quantified results as a new figure panel (Figure 2G) and a new supplementary figure (new, Figure 1—figure supplement 1 and new, Figure 3—figure supplement 1) and complemented the main text accordingly (paragraph three subsection “Cells exhibiting chiral counter-rotating actomyosin flows also undergo spindle skews”). Also refer to reviewer 2, point 2.

4) The authors use par mutants to assess the link between fate, chiral rotations, and spindle skew. As predicted, in par mutants that yield two AB like cells, both cells undergo counter-rotating flows and high levels of spindle skew, while mutants yielding two P_1_ like cells show no chiral rotations and reduced spindle skew. However, P_1_ cells have an active spindle orientation pathway which may dominate the system, preventing analysis of actomyosin generated spindle skew.

We thank the reviewer for bringing up the argument that the known spindle orientation pathway in the P_1_ cell might prevent an analysis of actomyosin generated spindle skew. Before we address this concern, we first want to point out that the mentioned spindle reorientation pathway in the P_1_ cell acts during early during mitosis (prophase) (Rose and Kemphues, 1998, Development), while the spindle skew events that we report occur much later, during anaphase. We have now modified the text to clarify this point results (subsection “Chiral counter-rotating ows drive spindle skews”).

In order to determine whether the spindle orientation pathway might prevent actomyosin generated spindle skew, we have perturbed the spindle orientation pathway in P_1_ by performing *lin-5* and *gpr-1/2(RNAi)* (Srinivasan et al., 2003). Upon knock-down of *lin-5,* we observed spindle defects that have been previously described, including a misaligned metaphase spindle in the P_1_ cell (new, Figure 3—figure supplement 2C, E) and confirming that the spindle orientation pathway was attenuated. Although the spindle was often misaligned, during anaphase we did not observe any spindle skew in P_1_ (new, Figure 3—figure supplement 2I), while spindle skews were still observed in AB cells (new, Figure 3—figure supplement 2B). Therefore, we conclude that the spindle orientation pathway in P_1_ does not mask actomyosin driven cell division skew.

We have added these results in a new supplementary figure (new, Figure 3—figure supplement 2) and adjusted the main text accordingly (subsection “Chiral counter-rotating ows drive spindle skews”).

I also worry about this analysis due to concerns in point 2 above – tilt in X-Y seems very different from spindle skew that would be driven by counter-rotation. Finally, it is unclear to me what kind of skew to expect when one obtains two adjacent cells with similar counter-rotations (e.g. Figure 2D), which I would have expected might just cancel out if divisions are all symmetric.

We thank the reviewer for bringing up this important point related to the direction of the spindle skew in 3 dimensions, and related to concern 2. Again, we agree with the notion that chiral flows are expected to tilt the division in the YZ-plane and that it seems not obvious how they could result in skew in the XY-plane. Similar to our reply to point 2, we have addressed this point by performing the posterior par RNAi perturbation in uncompressed embryos by embedding the embryo’s in low melt agarose. We first confirmed that the skew in the XY-plane is similar in compressed and uncompressed embryo’s (new Figure 2F and 2H) showing that our key results are not affected by embryo compression. In addition, we observed a clockwise skew in the YZ direction, similar to the AB cell in uncompressed wild type embryos (see response to point 2). Strikingly, when viewed from either embryo pole, the closest AB-like cell skews clockwise, meaning that the two juxtaposed cells counterrotate (new, Figure 3—figure supplement 2) and new Video 14–16.

Notably, the direction of rotation of both cells in this particular case can be explained by the cell-cell contact area having an increased friction coefficient when compared to the cellvitelline membrane and/or cell-eggshell contact. Given this topology, the handedness of the chiral flow indeed would give rise to clockwise rotation of both cells when viewed from the closest pole. These findings further substantiate our model in which chiral flows together with inhomogeneous friction drive cell skews.

As mentioned in our response to point 2, the exact direction of cell division skew in 3 dimensions would involve an in-depth analysis of surface flows and friction coefficients along the cell surface and is beyond the scope of this study.

To clarify these issues, we now provide a new supplementary figure (new, Figure 3—figure supplement 2) and three new Videos 11–13, we have adjusted the Results section accordingly and we have added a part to the Discussion.

5) The authors nicely show how counter-rotation depends on myosin activity (though this is known from their prior work and already explored extensively for Aba/p cells). Notably, at low levels of myosin, counter-rotation declines, which they then explore in Figure 3 and 4. However, the link to spindle skew is tenuous for the AB cell. Could one not explain the data more simply by imagining that cell length extension and the contractility of the cortex during cytokinesis combined with eggshell geometry drives tilt? A more compliant / soft cortex might simply deform during division and hence tilt less, while a stiff/contractile cortex would require the dividing cell to tilt as cell poles encounter the eggshell. This would also be compatible with the coordinated tilting of the two identical AB like cells produced by par mutants, which seems hard to explain from the balance of chiral flows alone.

We agree with the reviewer that we cannot fully exclude that cell length extension together with overall contractility of the cortex and eggshell geometry drives tilt. However, there are two observations that we believe argue against this hypothesis:

1) The force driving cell length extension is likely to be provided by the elongation of the mitotic spindle apparatus during anaphase. We have quantified spindle elongation in control conditions and upon *ect-2(RNAi)* (reduced chirality) and *rga-3(RNAi)* (increased chirality) and, although cell skews correlated with the chiral flow velocity, elongation of the mitotic spindle was similar in all three conditions (Supplement Figure 4A). These results rule out the possibility that spindle elongation, together with egg shell geometry drives the observed skew.

2) In order to determine the role of chirality in AB cell skews, our RNAi treatments of *ect-2* and *rga-3(RNAi)* were subtle and did not lead to a full knock-down of either gene product. Given that cytokinesis occurred normally under these conditions, we infer that overall contractility is not strongly affected. Corroborating this statement is our finding that the nonchiral component of the flow (the x-velocity, or contractile velocity) is not significantly affected in either RNAi (Figure 4C). In contrast, the chiral velocity is significantly affected and this correlates strongly with the skew extent (Figure 4B).

Although we cannot formally rule out the possibility that overall contractility, together with cell length extension and egg shell geometry is driving the AB skew, the tight correlation between chirality and cell division skew upon several different genetic perturbations strongly suggests that chirality, rather than overall contractility, drives the AB cell skew.

In order to clarify these issues, we have changed our wording in the main text.

6) The proposed model suggests that furrow position and myosin asymmetry are the key differentiating features of P lineage cells that prevent counter-rotating flow. However, the other obvious difference is that the two lineages show dramatic differences in actomyosin contractility and as the authors show myosin activity is directly related to the magnitude of counter-rotation. Can the authors exclude that reduced myosin activity in P lineage cells, rather than ring position and myosin asymmetry, is the key difference?

We thank the reviewer for this question and apologize if this point was not clear in the manuscript. While overall actomyosin contractility in the P_1_ lineage is lower than in the AB lineage, the actomyosin contractility during division of the P_0_ cell is not (new, Figure 1—figure supplement 5G). Still, this first cell division is asymmetric in terms of the ring position and myosin ratio, and it does not display any counter-rotating flows. This suggest that indeed the contractile ring position and myosin ratio, rather than the overall actomyosin contractility, are key determinants of counter-rotating flows.

We have adjusted the text for this concern and concern 7 in the Results and added a new Supplementary panel (new, Figure 1—figure supplement 5G) to improve clarity on this point (see also response to question 7b, from reviewer 2).

The model makes two key predictions that the authors do not test: First, a symmetrically-positioned cleavage furrow should lead to counter-rotating flow independent of myosin asymmetry – hence P_0_ cells with spindle positioning defects should exhibit counter-rotating flow.

We agree with the reviewer that a symmetrically positioned cleavage furrow would be expected to lead to counter-rotating flow independent of myosin asymmetry. In order to test this hypothesis, we performed a *lin-5 (RNAi)* treatment and analyzed cortical flows during the division of P_0_ cells. LIN-5 is involved in generating asymmetric pulling forces that displace the spindle from the center (Galli et al., 2011). We first confirmed that the myosin ratio in these P_0_ cells is still significantly different from one in these embryos (albeit reduced when compared with the control), while the ring position is centered (new, Figure 1—figure supplement 10A). Moreover, as the reviewer predicts, quantifying cortical flows revealed that counter-rotating actomyosin flows do emerge upon *lin-5 (RNAi)* treatment (new, Figure 1—figure supplement 10C). These results confirm our theoretical predictions and further substantiate our conclusions. However, we note that the myosin asymmetry is much less profound than in wild type cells. Therefore, it is difficult to fully decouple the effects of ring position and myosin asymmetry experimentally (see also response to question 4. from reviewer 1).

We have added these results in the new supplementary figure (new, Figure 1—figure supplement 10) and added the findings to the Results section.

Second, polarized, asymmetrically dividing cells such as P_0_ should never show chiral counter-rotation regardless of actomyosin levels. What happens in P_0_ if actomyosin activity is enhanced, e.g. with a strong rga-3/4 (RNAi), which the authors showed in 2014 enhances chiral flows in the one cell embryo?

We thank the reviewer for this suggestion and we agree with the statement that polarized, asymmetrically dividing cells, such as P_0_, should never show chiral counter rotation regardless of actomyosin levels. To test this, we first analyzed cortical flows upon an increase of RhoA signaling levels. Instead of using *rga-3(RNAi)*, we used an *ect-2* gain of function allele; *(ect-2 (gof)*) (Canevascini et al., EMBO rep 2005). While the myosin concentration increased when compared to control, we observed no chiral counter rotation in this gain of function allele (see Author response image 1). In addition, we also performed *par-2 RNA*i to make the ring position and the myosin ratio symmetric during the first cell division. In this condition, we do observe chiral counter rotating flows (new, Figure 1—figure supplement 9 C, D). Finally, these chiral counter rotating flows were dramatically enhanced in the *ect-2* gain of function background (see Author response image 1). Taken together, increasing RhoA signaling levels alone in asymmetrically dividing cells is not sufficient to induce chiral counter-rotating flows. These results are consistent with the model predictions and corroborate our finding that chiral counter-rotating flows can only emerge in symmetrically dividing cells.

We have added the analysis of the P_0_ division upon par-2(RNAi) as new, Figure 1—figure supplement 9 and added the findings to the Results section.

Our findings using the *ect-2* gain of function allele, we would prefer not to publish because this allele is central to an ongoing project in the lab. We have therefore added this data as a separate figure (see ) for the editors and the reviewers to consult. However, if the reviewers insist, we could include these results as a supplementary figure.

**Author response image 1. sa2fig1:** Enhancing actomyosin activity is not sufficient to generate chiral counterrotating flows during cytokinesis. A, B: Histogram of instantaneous chiral counter-rotating velocity vc for the first, P_0_ cell division in, control (A) and *ect-2 (gof)* embryos (B). C, D: Mean myosin concentration profile along the cell division axis of the P_0_ cell in control and *ect-2 (gof)* embryos. Vertical black line indicate centre of the cell. E, F: Histogram of instantaneous chiral counter-rotating velocity vc for the first cell division in par-2 (RNAi) embryos in control (E) and *ect-2 (gof)* embryos (F). Solid lines in A, B, E and F indicate the best fit Gaussian probability density function. Dotted vertical lines indicate mean vc; grey boxes represent the error of the mean at 95% confidence interval. G: Mean myosin concentration profile along the cell division axis of the first cell division in control, *par-2 (RNAi)* embryos. Vertical black line indicate centre of the cell. Right, normalized myosin concentration levels in the 20% anterior end of the cell compared to 20% of posterior end of the dividing cell. Shaded region indicates error at 95% confidence interval.

7) The authors make a strong point that given symmetric myosin, there is nevertheless a weak influence of furrow position. However, this plot (Figure1-Supplementary figure 3E) is somewhat misleading as the maximal effect on rotation is ~1% if myosin is symmetric – one could as easily argue that there is an insignificant contribution of furrow position relative to other factors.

The reviewer is correct that the amplitude of this effect as predicted by our minimal model is small, which can also be seen in Figure 1—figure supplement 7B. We believe that this is consistent with the experimental data, as it would effectively contribute the robustness of dominating counter-rotations during symmetric divisions, even if the furrow position is not perfectly symmetric. As mentioned in our response to question 4. from the first reviewer and in the first part of our response to the previous question 7, experimentally we could only partially decouple the role of contractile ring position and myosin asymmetry to confirm this hypothesis. However, our new results using *lin-5(RNAi)* (see response to reviewer 2 comment 7A) are in line with our theoretical predictions on the effect of ring position on chiral flows.

We clarified this point in the main text and adjusted our wording accordingly and as stated above have added our results using *lin-5(RNAi)* in new, Figure 1—figure supplement 10 and new, Figure 3—figure supplement 2.

[Editors’ note: what follows is the authors’ response to the second round of review.]

Essential revisions:The reviewers note that the motion of an elongated objects in a rotational flow is one of the best-studied problems in viscous fluid mechanics and conclude that even with the obvious complexities in the present setup (geometry, non-Newtonian fluids, etc.) it should be possible to make some sensible estimates.

We agree with the reviewers that it was not obvious how the tilt of the AB cell in the X-Y plane (or AP-DV plane) could be controlled by chiral counter-rotating flows, and we also agree that a lack of explanation for this particular skew was a weakness of the manuscript. Therefore, we have analyzed this particular skew event, and the accompanying chiral counter-rotating flows, in more detail, and found experimental as well as theoretical evidence that indeed strengthens our proposed hypothesis.

In our manuscript we propose that chiral counter-rotating flows, together with inhomogeneities in friction with the surroundings, will determine the direction of cell division skews. When viewed from the left side, the skew of the AB cell in the X-Y (AP-DV) plane is always clockwise. Given the handedness of the chiral counter-rotating flow, for this to occur, the friction forces experienced by the dividing AB cell must be higher on the right side than on the left side of the embryo. Given that friction forces act opposite to the direction of cortical flow, we measured cortical flow velocities in the AB cell on the future left and future right side. Note that previously, we have only presented data measured in the AB cell at the future right side (we only reported velocities on both sides for the PAR-perturbation experiments, Figure 3 B, C). We found that the velocities, and thus the v_c_, on the future left side, are significantly higher than on the future right side (new Figure 5—figure supplement 2). This is indeed consistent with the friction forces, being higher on the right side. Moreover, we have estimated the friction coefficient by fitting our active chiral fluid theory to the measured velocity distributions on the left and right sides. This yielded a friction coefficient that was higher on the right side than on the left side, again consistent with friction forces being higher on the right side of the AB cell. Finally, we estimated the cell-egg shell contact surface area from our confocal videos and found the right side of the AB cell having a slightly increased surface area that touches the egg shell. Again, this is consistent with the right side experiencing higher friction forces than the left side of the AB cell.

To further elaborate on these findings, we first derived a parameter-free geometric description that relates measured chiral flow velocities on the left and right sides of the AB cell to the cell division skew. In this model we discriminate between the cortical flow velocities in the lab frame (as measured directly from our videos) and the flow velocities in the frame that is co-rotating with the skewing cell. Here, the difference in cortical flow velocity in the lab frame and the co-rotating frame is due to the rotation of the system as a whole, e.g. the skew of the dividing AB cell. Using this simplified parameter-free geometric model the AB cell skew direction as well as its speed can be computed from the measured chiral flow velocities on the left and right sides. Indeed, predictions using this model correlate well with the experimentally measured cell division skew direction in the AP-DV plane, as well as its magnitude for wild type, and upon RhoA perturbations *(ect-2(RNAi) and rga-3(RNAi)).* We think that this is a large step forward and a big improvement to the manuscript which has come about by this comment, and we thank the reviewer for it.

In addition, we built a physical model that considers the torque balance of the AB cell with the eggshell. For this we explicitly considered friction forces with the eggshell on the left and right sides, by considering the areas of contact on the two sides and the respective frictional coefficients. Using the estimated friction coefficients as well the measured egg shell contact surface areas on the left and right side as input parameters, this physical model correctly predicts the direction of the AB cell skew in the AP-DV plane and qualitatively recapitulates the skew speed observed in wild type, and upon RhoA perturbations *(ect-2(RNAi) and rga-3(RNAi)).* Together, these findings underline quantitatively our proposed mechanism, indicating that the AB cell division skew in the X-Y (AP-DV) plane is the result of chiral counter-rotating actomyosin flows together with a left-right asymmetry of friction forces.

Altogether, our results are consistent with the right side of the AB cell experiencing higher friction forces with the egg shell than the left side. We also found indications that this is due to both an increased friction coefficient as well as an increased contact area. We note, that a previous study has shown that the cytokinetic ring of the AB cell ingresses in a left-right asymmetric fashion (Schonegg et al., 2014). Given that ring ingression likely decreases the contact area with the egg shell, these results are consistent with our new findings. We have modified the discussion of the paper where we further elaborate on these findings.

We are very thankful to the reviewers for pointing out this important issue. It has motivated us to look in more detail at the cell division skew in the AB cell. We feel that our additional experimental and theoretical analysis has substantially improved the manuscript.

We now provide two new theory predictions in Results section, two new supplementary figures and an appendix section (new Figure 5—figure supplement 2 and 3). Moreover, we have changed one paragraph in the Discussion to further elaborate on our new findings.

In this context, a particular statement that the reviewers struggled with was "Strikingly, when viewed from either embryo pole, the closest AB-like cell skews clockwise, meaning that the two juxtaposed cells counter- rotate (new, Figure 2—figure supplement 2) and new Video 14–16." The reviewers argued that the direction of rotation of both cells in this particular case can be explained by the cell-cell contact area having an increased friction coefficient when compared to the cell- vitelline membrane and/or cell-eggshell contact. Given this topology, the handedness of the chiral flow indeed would be expected to give rise to clockwise rotation of both cells when viewed from the closest pole." This raises the question of whether the cells counter-rotate or both rotate clockwise. The reviewers also note that the observed X-Y tilt of the cells in the same direction has not been addressed here.

We thank the reviewers for bringing it to our notice that this was not clear, and we agree that the term “counter-rotate” is confusing in this context. Moreover, we fully agree with the reviewer that the direction of the skew can be explained by flow handedness together with the cell-cell contact area having an increased friction coefficient when compared to the cell-vitelline layer and/or egg shell. The handedness of the flow is indeed predicted to lead to clockwise rotation of either cell when viewed from the closest pole. In addition, the reviewers correctly point out that the skew of both AB-like cells in the AP-DV plane in compressed embryos is occurring in the same direction. This is not immediately obvious from the handedness of the flow together with inhomogeneous friction forces. However, one of the two AB-like cells started skewing before the other one. Therefore, we argue that the AB-like cell that skews first results in geometry changes of the embryo that may push the neighboring cell, thereby biasing its cell division skew direction.

In order to clarify these points we have now changed the text in the Results section.

Finally, the notion emerged that the manuscript is not particularly reader-friendly, a concern that should be addressed. It is recommended to add a clear schematic early on that explains the geometry more clearly than Figure 1 (or the supplements).

We thank the reviewers for these comments and have added new clarifying illustrations in main Figure 3E and Figure 3F, that explain the geometry and the projection planes more clearly. In order to add these illustrations without decreasing figure panel sizes, we subdivided Figure 2 over two figures. Moreover, we have modified the text to make it more reader-friendly, especially when addressing the results using different projection planes. For example, we have changed our terminology and are now using the embryo body axes to describe projection planes. Hence, instead of the “X-Y plane” and “Y-Z plane” we now used “AP-DV (anteroposterior-dorsoventral) plane” and “LR-DV (left-right-dorsoventral) plane” consistently throughout the manuscript.

To clarify these issues, we have changed and rephrased many sections in the main results.

Moreover, the terminology "no chiral counter-rotating flows" was found to be confusing. Does that mean that there are chiral flows but they are not counter-rotating, or there are counter-rotating flows but they are not chiral? Indeed, chiral in this sense is not well-explained geometrically (here a schematic could help), especially since the experiments appear to only look at planar projections of the flows.

Our results show that there are chiral, counter-rotating flows in the AB lineage. In the P lineage, we observed flows that undergo net rotations, where both halves of the dividing cell move in the same direction. These net-rotating flows are therefore chiral, but are not counter-rotating. We have now explained these results more explicitly in the Results section. In addition, we have added a new illustration that clarifies the difference between chiral counter-rotating flows and chiral, net-rotating flows (new Figure 1—figure supplement 2 panels A and B), and we have now used the term “chiral counter-rotating flows” (instead of “chiral flows”) consistently throughout the manuscript.

To clarify these issues, we now provide two new subpanels in supplementary figure (new, Figure 1—figure supplement 2A – B)

We would also like to encourage you to include the species studied in the title.

We have added the species in the title of the study.